# Light-responsive self-strained organic semiconductor for large flexible OFET sensing array

Mingliang Li[1,10], Jing Zheng[1,10], Xiaoge Wang[2,10], Runze Yu[3,4], Yunteng Wang[5], Yi Qiu[2], Xiang Cheng[1], Guozhi Wang[6,7], Gang Chen[3,4], Kefeng Xie[8] ✉ & Jinyao Tang[1,9] ✉

With the wide application of organic semiconductors (OSCs), researchers are now grappling with a new challenge: design and synthesize OSCs materials with specific functions to satisfy the requirements of high-performance semiconductor devices. Strain engineering is an effective method to improve the semiconductor material's carrier mobility, which is fundamentally originated from the rearrangement of the atomic packing model of materials under mechanic stress. Here, we design and synthesize a new OSC material named AZO-BTBT-8 based on high-mobility benzo[b]benzo[4,5]thieno[2,3-d]thiophene (BTBT) as the semiconductor backbone. Octane is employed to increase molecular flexibility and solubility, and azobenzene at the other end of the BTBT backbone provides photoisomerization properties and structural balance. Notably, the AZO-BTBT-8 photoisomerization leads to lattice strain in thin-film devices, where exceptional device performance enhancement is realized. On this basis, a large-scale flexible organic field-effect transistor (OFET) device array is fabricated and realizes high-resolution UV imaging with reversible light response.

Strain engineering is an important method to modulate the physical properties in conventional inorganic semiconductor material. The strained silicon, germanium, GaAs, and newly developed 2D materials have been widely used and studied for high-performance electronic circuits[1–4]. Over the past decade, due to the potential application in flexible electronics, such as electronic skin and low-cost flexible displays[5–7], the strain effect on organic semiconductors is also extensively studied, where impressive material performance improvements are observed[8–10]. It's worth noticing that, due to differences in material structure, the strain effect shows fascinating distinctions in inorganic and organic semiconductors, which deserves further investigation. On the other hand, for photoresponsive molecules, if the photo-isomerization group is incorporated, molecules' rearranged packing can induce intrinsic strain, which has been utilized in light-responsive actuators[11–13]. It is natural to the hypothesis that the light-induced strain in organic semiconductors should also demonstrate light-strain modulation. To our knowledge, such a light-responsive self-strained organic semiconductor has not been proposed, although the such

[1]Department of Chemistry, The University of Hong Kong, Hong Kong 999077, China. [2]Beijing National Laboratory for Molecular Sciences, State Key Laboratory for Structural Chemistry of Unstable and Stable Species, College of Chemistry and Molecular Engineering, Peking University, Beijing 100871, P. R. China. [3]School of Physical Science and Technology, ShanghaiTech University, Shanghai 201210, China. [4]Shanghai Synchrotron Radiation Facility, Shanghai Institute of Applied Physics, Chinese Academy of Sciences, Shanghai 201204, China. [5]Institut für Geotechnik, Universität für Bodenkultur Wien, Feistmantelstraße 4, 1180 Vienna, Austria. [6]GRIMAT Engineering Institute Co., Ltd, Beijing 101407, P. R. China. [7]State Key Laboratory of Advanced Materials for Smart Sensing, General Research Institute for Nonferrous Metals, Beijing 100088, P. R. China. [8]School of Chemistry and Chemical Engineering, Lanzhou Jiaotong University, Lanzhou 730070, China. [9]State Key Laboratory of Synthetic Chemistry, The University of Hong Kong, Hong Kong 999077, China. [10]These authors contributed equally: Mingliang Li, Jing Zheng, Xiaoge Wang. ✉e-mail: xiekefeng@mail.lzjtu.cn; jinyao@hku.hk

effect may have been unintentionally used in many previous photo-responsive devices. Here, we developed a strategy-directly grafting the photochromic groups onto the high-performance semiconductor molecule motif and studied the light-induced self-straining effect. Specifically, benzo[*b*]benzo[4,5]thieno[2,3-*d*]thiophene (BTBT) is employed as the semiconductor backbone due to its high intrinsic mobility[14]. The BTBT backbone is alkylated on one end with octane to improve molecular flexibility and solubility[15], while the opposite end is covalently bound to azobenzene (AZO) to endow with reversible photoisomerization property[16,17]. These molecules with asymmetric modification can also obtain good molecular packing through simple post-processing, such as thermal annealing, thus improving performance[15,18,19]. The AZO group in the as-synthesized AZO-BTBT-8 undergoes a switch from *trans* to *cis* conformation upon ultraviolet (UV) irradiation and back to the thermodynamically stable *trans* conformation with visible-light (Vis) excitation or high-temperature treatment[20,21]. Due to the steric hindrance[16,22–24], most photo-inducing folding occurs on the top thin layer, in contrast to a fraction of the molecules inside the OSCs film. The vertical layer difference will induce uniform lattice strain to the bulk semiconductors, thus positively feedback to long-range ordered crystalline[25,26] and increasing the mobility. On this basis, a large-scale flexible OFET device array is fabricated utilizing AZO-BTBT-8 as the active semiconductor layer, exhibiting reversible light response and good stability under complex deformation, further indicating the AZO-BTBT-8 molecule performs an experimental foundation for molecular engineering and strategy optimization with specialized functionality.

## Results and discussion
### Characterizations of AZO-BTBT-8

As designed, the molecular structure, DFT-calculated molecular conformation, and frontier orbitals of AZO-BTBT-8 are shown in Fig. 1a. AZO-BTBT-8 mainly presents as planar *trans* conformation in the ambient environment, while the UV light triggered the isomerization to *cis* conformation with benzene rings tilted to each other. This photoisomerization slightly increases the highest occupied molecular orbital (HOMO), benefiting the energy level alignment with electrode Au (−5.1 eV) in the top-contact/bottom-gate device architecture (Fig. 1b).

All the experiments are conducted with spinning-coated film samples annealed at 80°C unless otherwise stated. The thermodynamic properties of the bulk material were first investigated. Supplementary Fig. 2a shows the breakdown temperature of 340°C,

demonstrating strong thermal stability, which is a critical requirement for OFET electronics. According to the differential scanning calorimetry plots (DSC, Supplementary Fig. 2b), liquid crystals and Maltese crosses were revealed by polarized optical microscope (POM) in the incubation at 80°C (Supplementary Figs. 3 and 4). Thus, to optimize molecular packing in thin-film devices, an annealing temperature of 80 °C (30 min, Supplementary Fig. 5) was selected to help with the molecular rearrangement[27,28]. AZO-BTBT-8 thin film was further examined with atomic force microscope (AFM). As shown in Fig. 2a, the film of AZO-BTBT-8 shows good continuity and flatness with molecular step edges, which is suitable for electronic device fabrication. After exposure to UV light (365 nm, 20 mW•cm⁻²) for two hours, the average step height of the surface monolayer decreased from 3.11 to 2.85 nm (Fig. 2b), which is consistent with the DFT calculated value (27.2 Å to 24.9 Å, difference of 2.3 Å, Supplementary Fig. 16) and indicates the complete photoisomerization on the film surface. The time-dependent UV-Vis experiment is then conducted to investigate the photoisomerization kinetics, showing the reversible photoisomerization of AZO-BTBT-8 in both solid film and solution (Supplementary Figs. 9a and 10). However, compared to the solution (Supplementary Fig. 9b), the thin films sample (800 nm, Fig. 2d) shows a much weaker response, which can be attributed to the steric hindrance in the condensed phase[29]. Due to the steric hindrance, a fraction of the molecules is triggered for photoisomerization inside the film under UV irradiation, in contrast to almost 100% on the surface. Therefore, the heterogeneous conformational change in the film is predictable, with higher conversion ratio towards the top surface (Fig. 2c). As a result, the thicker film shows lower overall conversion ratio, slower conversion kinetics and longer half-life time ($t_{1/2}$), as observed experimentally in Fig. 2e and Supplementary Table 4. It is well-known that strain engineering is a general strategy applied in semiconductor materials to enhance device performance[30,31]. Previously, the strain-enhanced mobility in organic semiconductors has been achieved by mechanically shearing the growth solution during the crystallization, where a substantial mobility improvement is achieved[8]. In our thin-film device, since the top surface contributes to the majority of the photo-isomerization while the bottom surface contributes the majority to the electrical conductivity, the top surface isomerization can induce uniform lattice strain to the bulk semiconductors (Fig. 2c). Here, the molecule configuration deep inside the film is analyzed with grazing incidence X-ray diffraction (GIXD), conventional X-ray diffraction (XRD), and the electric property is monitored with conductive AFM. Contrary to the high crystallinity on the film surface, from the GIXD,

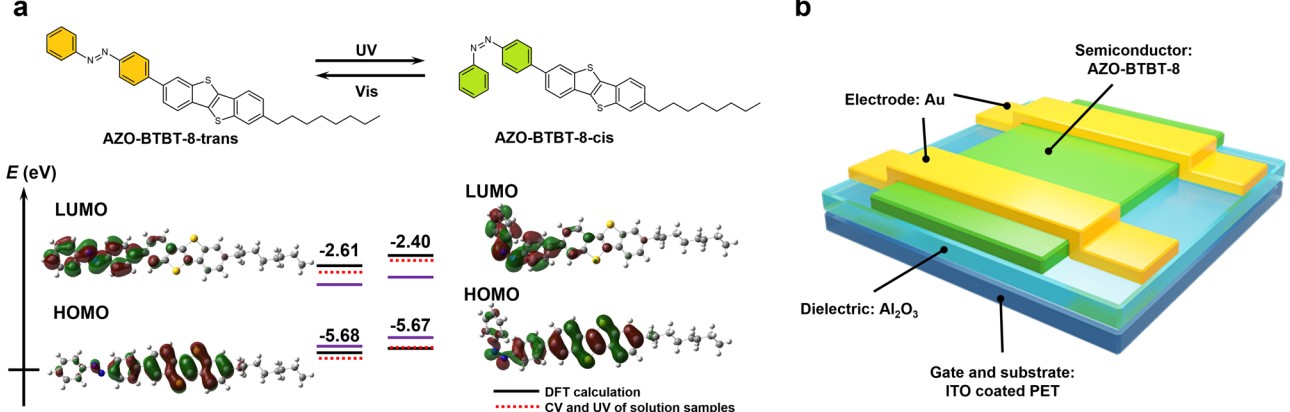

**Fig. 1 | Organic semiconductor AZO-BTBT-8 with photoisomerization property.** **a** Energy level diagram of *trans* and *cis* conformations of AZO-BTBT-8. The orbital values are obtained from DFT calculations (black solid line, Supplementary Table 2), CV (Supplementary Fig. 6), and UV−Vis absorptions measurements (red dashed line for solution samples and purple solid line for solid films, Supplementary Table 3). UV, ultraviolet (λ = 365 nm); Vis, visible (λ ≥ 420 nm). **b**, Schematic cross-section of an OFET with AZO-BTBT-8 as the semiconductor layer.

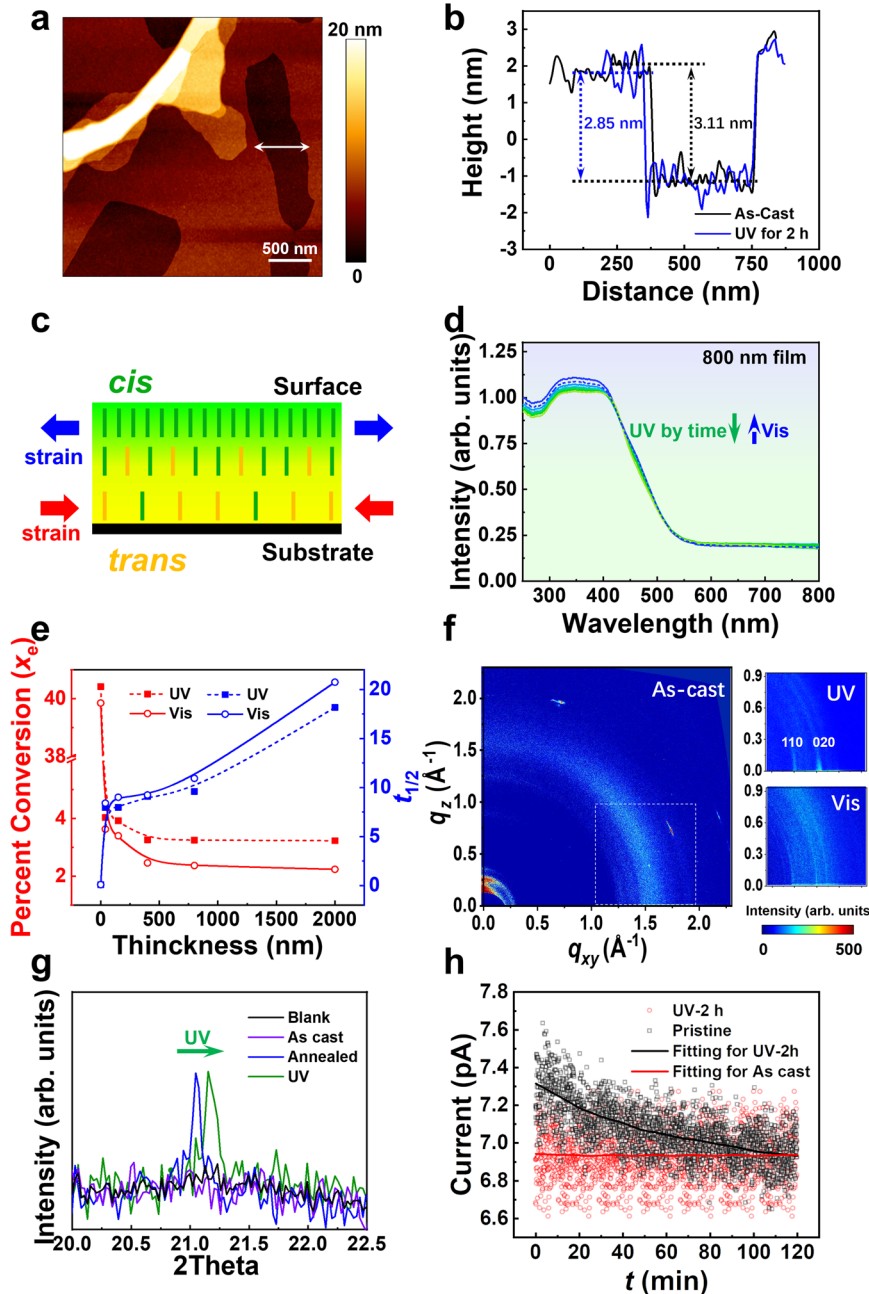

**Fig. 2 | Photoisomerization characterization of AZO-BTBT-8 films. a** The AFM image of AZO-BTBT-8 film. **b** In-situ height profile curves indicated by the white two-way arrow in **a**. **c** Schematic diagram for the distribution of molecular conformation and strain in AZO-BTBT-8 film after UV irradiation. Green and yellow lines represent the molecules with *cis* and *trans* conformations, respectively. **d** Time-dependent UV-Vis spectra of an 800 nm film under ordinal UV and Vis irradiation. Green arrows represent the variation tendency of UV irradiation, and dashed blue arrows represent the variation tendency of visible light irradiation. **e** UV-Vis absorption and half-life time ($t_{1/2}$) investigation with film thickness. The thickness of the solution sample is set as 0 nm. Solution concentration for UV test is $10^{-5}$ M in chloroform. **f** GIXD diffraction pattern sequentially for the pristine (thermal-annealed), UV-irradiated, and Vis-irradiated samples. **g** Enlarged XRD patterns with 2theta from 20.0° to 22.5° from Supplementary Fig. 13. **h** Current decay fitting curve of conductive AFM. All the experiments are conducted with film samples annealed at 80 °C unless otherwise stated.

the film shows low-order diffraction ring (left of Fig. 2f), which indicates more irregular packing inside the bulk phase of the film. Upon UV irradiation, GIXD diffraction shifts into distinct Bragg rods (110) and (020), indicating improved packing in *xy* plane (upper right of Fig. 2f). Interestingly, after visible light irradiation, this pseudo crystalline phase can be switched back to lower-order phase with weak diffraction and a *d*-space expansion is observed, where (020) peak moves to higher fields with *d*-value increasing from 2.704 to 2.729 Å (lower right of Fig. 2f). It is well known that the strain can induce ordering and crystalline in amorphous materials, and the rich dynamics of glass

composed of photoisomerization molecules have been discussed recently[32]. Since this reversible crystallinity change can not be explained by direct photoisomerization of molecules, we attribute this effect to the strain-induced crystallization, while the details mechanism requires further investigation. From the thin film XRD, the *d*-space shrinking is also observed, where the diffraction peaks at 21.05° shift to higher fields under UV irradiation (Fig. 2g and Supplementary Fig. 13), with *d*-value of the main peak decreasing correspondingly from 4.093 to 4.073 Å. The photoisomerization-induced strain can change the carrier mobility and material conductivity, similarly to previous

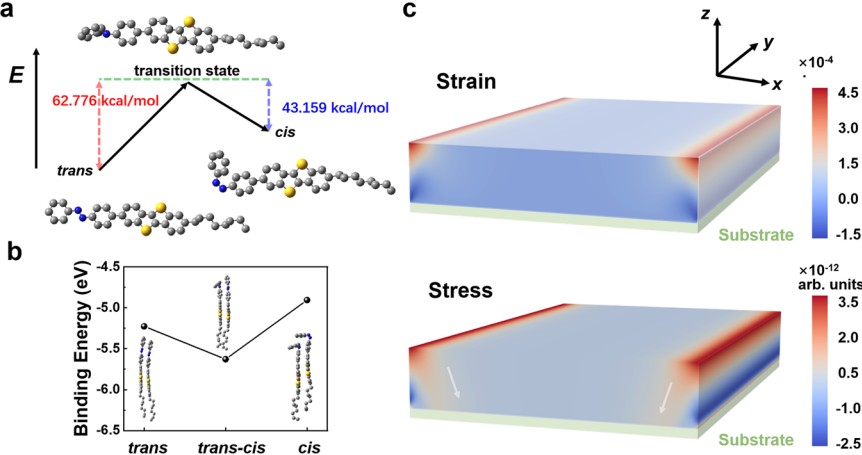

**Fig. 3 | Theoretical calculations of DFT and film mechanics. a** DFT energy schematic diagram under UV irradiation. *E* indicates molecular energy. **b** DFT molecular binding energy diagram. The insets are the corresponding packing structures. **c** The calculated distributions of the strain and stress in the long-axis direction (*x*-axis) in the organic film. The white arrow indicates the direction of stress decay. The aspect ratio of the film section is simplified to 8000:800. Stress, the internal forces that neighbour particles of a continuous material exert on each other. Strain, the measure of material deformation. Both stress and strain are calculated in the *x* dimension.

mechanical-induced strain[33]. As shown in Fig. 2h and Supplementary Fig. 8, the conductive AFM scan shows a significant and instant current increase upon UV irradiation and a gradual current decay ($K = 0.01587$ min$^{-1}$, ca. 120 min falling with the baseline) at room temperature in the dark, which can be attributed to the thermal isomerization of *cis* to *trans* in the film and the strain-induced ordering, resulting in high carrier mobility[25,34,35].

## Theoretical calculations

Moreover, it is calculated that *cis* is an unstable conformation excited from *trans* (Fig. 3a). However, compared with those with single conformation, the binding energy between *cis* and *trans* is much higher (Fig. 3b), resulting in regular molecular arrangement and alignment, which also explains that despite the limited concentration of *cis*, especially inside the film, it is still able to optimize the molecular packing and further stress on the deeper molecules. In addition, the average distance between the *trans-trans* conformations (3.129 Å) in the DFT calculation is larger than that between *trans-cis* conformations (3.104 Å), which indicates light-induced isomerization also synergistically optimizes molecular interaction with dense stacking and ultimately improves device performance (Supplementary Fig. 17). With displacement boundary conditions, we simplified the film section and calculated the strain and stress distributions (Fig. 3c and Supplementary Fig. 18). It was observed that after UV irradiation both the stress and strain accumulated along the *x*-axis and eventually released at the film edge. The stress gradually decreases with film depth and ultimately reaches the interface between dielectric and semiconductor, affecting the device output through molecular packing optimization. This is in good agreement with the above-proposed mechanism.

## Device properties

The photo-switchable OFET devices were then fabricated based on a bottom-gate/top-contact architecture (Fig. 1b) to characterize the electrical properties and mechanical stability. The cross-sectional scanning electron microscope (SEM) and element mapping show the homogeneous layered structure of the architecture, which is essential to minimize the macroscopic carrier hindrance (Supplementary Fig. 14). Furthermore, after the deposition of gold electrodes, a new fraction of the sulfur peak appears to the lower-field in the XPS plots (Supplementary Fig. 15) due to the development of a high-strength S-Au bond[27], which can reduce the contact barrier between the semiconductor and the electrode. Then the electrical characteristics were examined to quantify the photoisomerization (Fig. 4, Supplementary Figs. 19 and 20). Compared with the negligible photocurrent under UV irradiation (Supplementary Fig. 21), the carrier mobility of annealed device on PET rises from $0.015 \pm 0.007$ cm$^2$ V$^{-1}$ s$^{-1}$ (Fig. 4a–b) to $0.141 \pm 0.009$ cm$^2$ V$^{-1}$ s$^{-1}$ (Fig. 4c–d) with a fixed on/off ratio of $10^5$, showing an increase of 9.4 times on average due to optimized molecular stacking as described previously. In comparison to the device on silicon (Supplementary Fig. 22 and Supplementary Table 6), the flexible device maintains the expected response with moderate performance loss for the rough and bending substrate (Supplementary Table 1). Interestingly, the mobility of as-cast film is also improved after UV irradiation, which indicates that the strain effect generated by photo-isomerization packs the molecules more regularly, similarly to thermal annealing. Furthermore, it is found that the device photo-response increases with UV intensity at the low intensity range (Supplementary Fig. 23). Intense irradiation, on the other hand, may result in molecular degradation, against the device performance. The relationship between incident angle and light response has also been investigated (Supplementary Fig. 24). Due to the varied optical route, the device response is observed to be unaffected by the constant radiant intensity on the top surface and the various intensities on the bottom interface. On the other hand, front and back irradiation produce different responses to intense (20 mW cm$^{-2}$, enough light intensity to penetrate the device) and week (100 μW cm$^{-2}$, not enough intensity to penetrate the device) UV light, respectively. Both of these results suggest that the photoisomerization of the top surface has a greater impact on the OFET device's photoresponse. To estimate the folding capability, a homemade apparatus was designed to realize controllable convex and concave bending (Supplementary Fig. 25). As the PET film is thin, the tensile and compressive stress were both small in the bending test, and the devices showed negligible performance deterioration during the deformation (Fig. 4e). All the devices showed steady device performance in the deformation test with strain, with negligible degradation of the transfer curve (Fig. 4e inset). It is noted that the device also exhibited excellent cyclic performance, its mobility and switching ratio stayed constant at 0.015 cm$^2$ V$^{-1}$ s$^{-1}$ and $10^5$ after 200 bending cycles, respectively (Fig. 4f), and the drain current was reversible under alternant UV and visible light illumination. As shown in Fig. 4g, the light-inducing isomerization caused a current change with a stable magnitude of around 10 times ($V_{DS} = -25$ V, $V_G = -25$ V). On the other hand, heating also leads to *cis-trans* isomerization. As

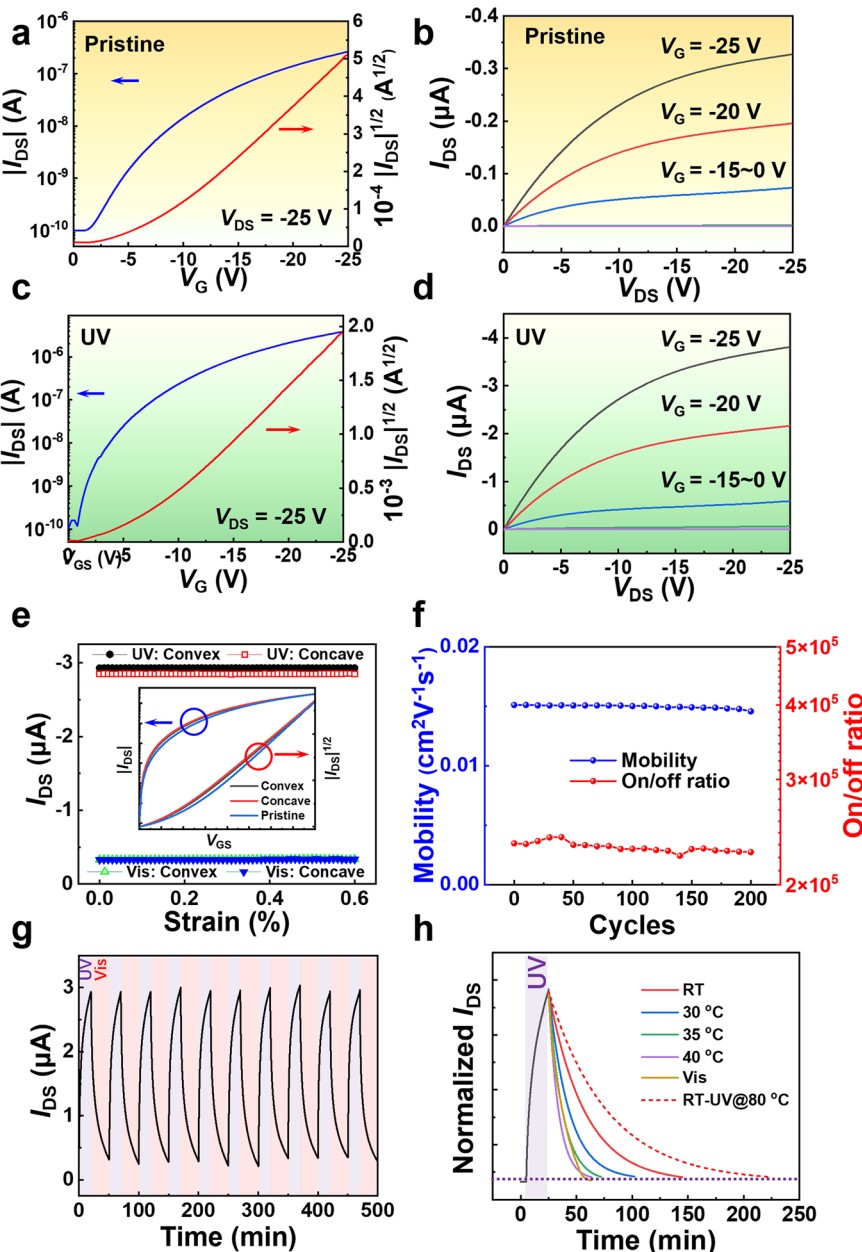

**Fig. 4 | OFET device performance. a–d** are the transfer and output curves of AZO-BTBT-8 before and after UV irradiation on flexible PET, respectively. $I_{DS}$, source-drain current; $V_{DS}$, source-drain voltage; $V_G$, gate voltage. The average mobility increased from $0.015 \pm 0.007$ cm$^2$ V$^{-1}$ s$^{-1}$ (**a** and **b** without UV irradiation) to $0.141 \pm 0.009$ cm$^2$ V$^{-1}$ s$^{-1}$ (**c** and **d** with UV irradiation) on over 30 individual devices. The average on/off ratio is approximately$10^5$. $L = 30$ μm and $W = 130$ μm. **e** $I_{DS}$ of the OFET device under various mechanical distortions. The inset shows the transfer curves of the same device under different bending conditions. *S*, strain calculated according to Supplementary equation 7 and 8. **f** Endurance cycles show the device's stability before UV irradiation on flexible PET. **g** Time trace of $I_{DS}$ for the same device showing the reversible photoswitching events under alternating UV and Vis irradiation. **h** Comparison of $I_{DS}$ attenuation at different temperatures and under Vis irradiation. $V_{DS} = -25$ V, $V_G = -25$ V. UV = ultraviolet ($\lambda = 365$ nm), Vis = visible ($\lambda \geq 420$ nm).

shown in Fig. 4h and Supplementary Table 5, the rate constant (*k*) of *cis* to *trans* conformation increased by heating, producing apparent activation energy of 65.67 kJ mol$^{-1}$, while the rate constant at 35 °C ($k = 0.07926$ min$^{-1}$) is comparable to that under Vis irradiation. Moreover, the device showed better mobility (Supplementary Fig. 26) and slower decay plot (dashed red line in Fig. 4h) at room temperature after UV irradiation at 80 °C, proving the important evidence of steric hindrance in solid-state photoisomerization.

### Sensing tests by the OFET array

The as-prepared single OFET device exhibits good performance as described above, however, to implement true high availability, how to make large-scale flexible array is inevitable[36,37]. To realize it, we design a strategy by depositing patterned electrodes directly on a desired flexible substrate, as shown in Fig. 5a. Specifically, patterned AZO-BTBT-8, source electrodes, hafnium oxide dielectric layer, and drain electrodes were deposited sequentially on an Al$_2$O$_3$ covered ITO-PET. Thus, the precise device array layout was obtained (Fig. 5b). Figures 5c and d shows the optical image of a 33 × 40 device array, the magnified view of the 3 × 3 device array, and the SEM image of the individual device, respectively. The average mobility increased from $0.016 \pm 0.005$ cm$^2$ V$^{-1}$ s$^{-1}$ to $0.152 \pm 0.009$ cm$^2$ V$^{-1}$ s$^{-1}$ under UV illumination with the on/off ratio of $10^5$ ($L = 50$ and $W = 1080$ μm), which is consistent with the individual OFET device and demonstrates the

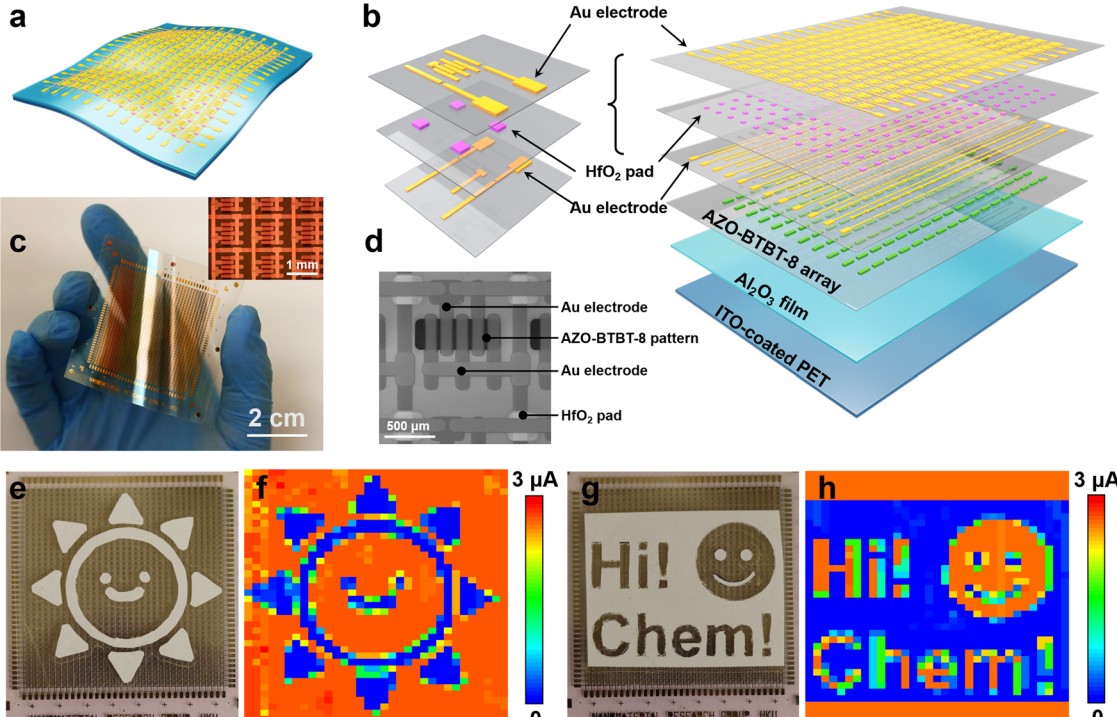

**Fig. 5 | Sensing performance with a large-scale flexible OFET array. a** Schematic diagram of an OFET array. **b** Layered structures of the OFET array from bottom to top: ITO-coated PET, 200 nm thick $Al_2O_3$ film, 100 nm AZO-BTBT-8 film, Au electrode, 100 nm thick $HfO_2$ square array, and Au electrode. The inset shows the magnified structure of the top three layers in one pixel. **c** The front-view photograph of a complete 40-by-33 OFET array on a flexible substrate. The inset shows a magnified optical image of the array. **d** The detailed SEM image of an individual pixel. **e–h** Top-view photographs and corresponding current mappings of a smiling cartoon sun (garland) and words "Hi! Chem!" with an emoji (diaglyph). $V_{DS}$ = -15 V, $V_G$ = -15 V, UV = ultraviolet ($\lambda$ = 365 nm, 100 $\mu$W cm$^{-2}$), Vis = visible ($\lambda \geq$ 420 nm). The device array is prepared by continuous evaporation of AZO-BTBT-8, Au and $HfO_2$ insulating layers by a set of aligned shadow masks without annealing.

reproducibility of the device performance. The optical pattern is obtained after top UV illumination to the array through a smiling cartoon sun garland mask for 20 minutes (Fig. 5e), where the exposed area switched from OFF state (-0.2 $\mu$A) to ON state (-2.8 $\mu$A) (Fig. 5f). Since the photoisomerization is reversible, the image can be erased with visible light irradiation and repatterned. As shown in Fig. 5g, h the previous pattern is erased with 30 min exposure of visible light and can be repatterned under the UV illumination with the diaglyph pattern "Hi! Chem!" and a happy emoji. All evidence suggests that the OFET array devices are reversible, programable, and scalable, with promising potential in other fields, like flexible sensors, displays, wearable devices, health care, and tissue detection.

In this work, a functional molecule AZO-BTBT-8 was designed and synthesized by integrating photochromic azobenzene, high-performance semiconductor backbone BTBT, and flexible alkyl chains. The light-induced self-strain engineering is observed in this material and leads to reversible mobility switching in solid-state devices. Based on this mechanism, a large-scale flexible OFET device was fabricated on a flexible substrate using AZO-BTBT-8 molecules as photoisomerization OSCs, showing good stability and reproducibility. In conclusion, this work establishes a new strategy for designing and developing light-responsive OSCs and corresponding functionalized OFETs.

## Methods
### Materials
All reagents and chemicals were obtained from commercial sources and used without further purification unless otherwise noted. All reactions were performed under an inert atmosphere of argon in dry solvents using standard Schlenk techniques. The synthetic route used to obtain linker AZO-BTBT-8 is outlined in Supplementary Fig. 1.

### Characterization
To ensure the reliability of the experiments, unless otherwise stated, all film spinning-coated samples are annealed at 80 °C for 30 min before being characterized and tested. The morphology of thin films was investigated by a JPK atomic force microscope (AFM) under ambient conditions in QI mode. The conductive AFM was conducted in contact conductive module. Film and powder X–ray diffraction data were collected on PANalytical high resolution PXRD. GIXD data were obtained at beamline BL14B1 of the SSRF at a wavelength of 1.2398 Å.

### Device fabrication and measurement
Heavily doped *n*-type silicon wafers were cleaned in a Piranha solution (volume ratio of components $H_2SO_4/H_2O_2$ = 70:30) by heating at 110 °C for 2 h followed by rinsing thoroughly with de-ionized (DI) water, sonicated for 15 min in an RCA solution (volume ratio of components DI water/ammonium hydroxide/$H_2O_2$ = 5:1:1), rinsed and dried under nitrogen, and used immediately. 200 nm $Al_2O_3$ thin film was deposited on the clean silicon wafer by atomic layer deposition (ALD) at 200 °C for 1850 cycles. A cleaned ITO-coated PET substrate (Sigma-Aldrich, 1 ft × 1 ft × 5 mil) was used as the flexible substrate. 200 nm $Al_2O_3$ thin film was deposited on PET by ALD at 70 °C for 2000 cycles. The device was fabricated by the spin-coating of a solution of AZO-BTBT-8 in 10 mg/mL CHCl$_3$ (annealed at 80 °C for 30 min) and thermal evaporation of Au through a designed mask. The device array on PET is prepared by continuous evaporation of AZO-BTBT-8, Au and $HfO_2$ insulating layers by a set of aligned shadow masks. The transistor characteristics were obtained at the room temperature in air by a standard probe station and two semiconducting parameter analyzers (Keithley 2400). The mobilities of the devices were calculated in

the saturation regime by the standard method:

$$I_{DS} = (W/2L)C_i u(V_G - V_T)^2 \qquad (1)$$

Where $W/L$ is the channel width/length, and $V_G$ and $V_T$ are the gate voltage and threshold voltage, respectively. $C_i$ is the insulator capacitance per unit area.

## DFT and mechanical calculations

**DFT calculation.** All geometric structure calculation has been carried out using Gaussian 09 package and Gauss view molecular visualizing program package which has provide itself to be extremely useful to get a clear knowledge of optimized parameters, electronic structure properties and other molecular properties. The geometry is fully optimized at Beck3-Lee-Yang-Parr (B3LYP) [1,2] level with standard 6-311 + G (d, p) basis set [38,39].

**Mechanical analysis.** The mechanical analysis model uses the displacement boundary condition, and it is assumed that all molecules in the film are stacked along the $x$-dimension, that is, all film deformations occur in the $x$- dimension. The aspect ratio of the film section is simplified to 8000:800. The Poisson's ratio and elastic modulus are estimated to be 0.35 and 7 GPa, respectively, according to the literature [40,41]. The strong form in the proposed thermodynamically consistent phase field model can be written as follows [42,43]:

$$\nabla \cdot \boldsymbol{\sigma}(\boldsymbol{\varepsilon},d) + \mathbf{b} = 0 \text{ in } \Omega \qquad (2)$$

$$\boldsymbol{\sigma}(\boldsymbol{\varepsilon},d) \cdot \mathbf{n} = \bar{\mathbf{t}} \text{ on } \partial\Omega_t \qquad (3)$$

$$u = \bar{u} \text{ on } \partial\Omega_u \qquad (4)$$

$$(1-d)\left(\frac{H_n}{G_{Ic}} + \frac{H_t}{G_{IIc}}\right)\eta + \frac{l_c}{2}\nabla^2 d - \frac{d}{2l_c} = 0 \text{ in } \Omega \qquad (5)$$

$$\nabla d \cdot \mathbf{n} = 0 \text{ on } \partial\Omega \cup \Gamma \qquad (6)$$

where $\mathbf{b}$ is the body force vector, $\bar{\mathbf{t}}$ is the traction vector, $u$ is the displacement field, $d$ is the phase field, $\boldsymbol{\sigma}$ is the Cauchy stress tensor, $\boldsymbol{\varepsilon}$ is the strain tensor, $G_{Ic}$ and $G_{IIc}$ are the critical fracture energy release rates for mode-I and mode-II fracture modes, respectively.

## Data availability

All the data generated or analyzed during this study are included in this published article (and its Supplementary Information files) or available from the authors upon request.

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

## Acknowledgements

This work was supported in part by the Innovation and Technology Commission (HKSAR, China) to the State Key Laboratory of Synthetic Chemistry, the Hong Kong Research Grants Council (RGC) General Research Fund (GRF17305917, GRF17304618, J.T.), Collaborative Research Fund (C7045–19EF, C7018–20G, C7082–21G, J.T.), RGC Research Fellowship (RFS2122-7S06, J.T.), the Seed Funding for Strategic Interdisciplinary Research Scheme (University of Hong Kong), Hong Kong Quantum AI Lab Ltd. (J.T.), Science and Technology Commission of Shanghai Municipality (YDZX20203100001438, G.C.), and the Shenzhen-Hong Kong Innovation Circle Program (SGDX2019081623341332, J.T.).

## Author contributions

J.T. and M.L. conceived and designed the experiments. M.L. performed the material synthesis and most of the device characterizations. J.Z. fabricated the OFET and device array. X.W. and Y.Q. did the XRD and related analysis. Y.W. performed the mechanical analysis. X.C. performed the SEM tests. R.Y. and G.C. conduct the GIXD tests. G. W. did the liquid crystal related tests. K.X. did the DFT calculation. M.L., J.Z., X.W., K.X. and J.T. analyzed the data and wrote the paper. All the authors discussed the results and commented on the manuscript. All authors have given approval to the final version of the manuscript.

## Competing interests

The authors declare no competing interests.
