## [Peer Review File · Nature Communications]

Light-responsive Self-Strained Organic Semiconductor for Large Flexible OFET Sensing ArrayEditorial Note: Parts of this Peer Review File have been redacted as indicated to remove third-party material where no permission to publish could be obtained.

REVIEWER COMMENTS

Reviewer #1 (Remarks to the Author):

This article describes a photo-responsive flexible OFET array based on an organic semiconductor containing a photo-isomerizable azo group and its device performance. Before judging quality of this paper, the reviewer felt difficulty in understanding the experimental results exactly because a condition for each experiment is not clearly described: for example, the reviewer wonders whether the samples after UV-irradiation were thermally annealed, which is very important to know what the new findings are in this paper. Therefore, the reviewer requests exact description particularly on the experimental condition including the main text and figure captions before describing review report.

Reviewer #2 (Remarks to the Author):

I have read the manuscript "Light-responsive Self-Strained Organic Semiconductor for Large Flexible OFET Sensing Array" by Mingliang Li et al. (MS # NCOMMS-22-04735) submitted for the publication in Nature Communications.

In their manuscript the authors reported the synthesis and characterization of AZO-BTBT-8 organic semiconductor (OSC), which was applied in the fabrication of a flexible OFET device with reversible response upon UV-Vis irradiation. Reversibility is gained thanks to the trans-cis isomerization of azo-pendants, which induces a reversible stress-strain deformation with a consequent increase of charge mobility. The claimed possibility of developing light responsive OSC was confirmed by the fabrication of a prototype with different masks. However, it is opinion of the referee that the most delicate point of the research design is if the small changes, induced upon irradiation, in the molecular packing could be the real cause of the observed effect (self-strain). As the same authors wrote, the effect should be limited to top of the thin layer (line 61), and:

1. the change in the height is low (Figure 2.b),
2. the shift in XRD peaks is very small (Figure 2.h),
3. and the current data show a large dispersion (Figure 2.i).

It is well known that the conductivity can change upon photoisomerization of azocompounds present in polymeric materials and can be used for the preparation of light-responsive large area devices. Consequently, before the publication in NatCom the authors should investigate a blank device, in which the AZO-BTBT-8 is not aligned (in such a way there should not be any evident self-strain) and verify the presence/absence of any effect.

In addition, in the manuscript should be present an adequate relevance of the liquid crystallinity of AZO-BTBT-8 and a deeper investigation (why there are only few Maltese crosses? Pictures samples with a planar alignment at different angle between polarizers are required, X-ray data for the mesophase, ...).

In addition:

1. Scheme 1: The reported synthesis route of AZO-BTBT-8 is not accurate;
2. SI Lines 152-154: please, check calculations for the 1st kinetics;
3. SI Line 161: the reported Xe equation gives the complement of the conversion from trans to cis and $\times 100$ is missing;
4. Line 101: from Figure 2.a the surfaces does not look like continuous and flat;
5. Please, check values for mobility reported in the text and in the Figure captions. In Figure 4.f are the reported mobilitites before or after UV irradiation?

Minor comments:

1. There some typos both in the manuscript and SI: thoroughly, 1 ft × 1 ft (these are the dimensions of the original PET substrate), Gpa, thermodynamically, scanning, tempertures,...;
2. SI line 124: data cannot agree with literature if AZO-BTBT-8 is a new compound;
3. Caption of Figure S2: it is not clear the means of "growing direction". Are the two transition temperatures really referred to crystal \leftrightarrow nematic \leftrightarrow isotropic phase transitions? Could be present a smectic arrangement at lower temperatures?
4. Some acronyms are not defined;
5. Caption of Figure S7: there are too many plots than absorption wavelengths;
6. What about the differences in absorption between plots in Figures S6 and S7?

Reviewer #3 (Remarks to the Author):

Review comments

The authors showed a very nice example where light-induced isomerization of a photochromic azobenzene based organic semiconductor (OSC) led to exceptional device performance enhancement and reversible mobility switching in solid-state devices. A large-scale flexible OFET device was also fabricated on a flexible substrate using such azobenzene as photoisomerization OSCs, showing good stability and reproducibility. Various characterizations of such phenomena are presented to draw a conclusion that such light-responsive self-strained OSC provides a new strategy for designing and developing light-responsive OSCs and corresponding functionalized OFETs. The results are very impressive, especially when it was realized in a large scale flexible array that are reversible, programable, and scalable.

The results are original and should be of broad interest to readers in the field and related fields. I would recommend acceptance after minor revision.

In the meantime, I suggest that the authors should also consider the changes in molecular packing caused by the light-induced isomerization as one of the reasons for the electronic performance enhancement. The height decrease indicates a closer packing, which usually leads to better mobility for OSCs. How does the intermolecular packing distance changes according to the calculations? The results on GIXD diffraction pattern indicate more crystalline packing. Whether that's a direct result of strain, or a strong intermolecular interaction, or the combination of both factors needs more discussion.

In addition, the following minor issues should also be corrected.

1. Please specify the digital numbers in Fig. 1a correspond to which energy level (DFT calculation, CV and UV of solution or the solid-state films?) in the figure caption.
2. Scale bar in Fig. 2a didn't seem to be right. Please double check.
3. The right axis $t_{1/2}$ in Fig. 2f was not mentioned or discussed in the main text or the figure caption.
4. Keyword "Calculated" should be added to the figure caption of Fig. 3c. And description on calculation methods, the aspect ratio of the film section should also be included in figure caption.

Listed below are the details of our responses to the referees' comments.

Reviewer #1:

This article describes a photo-responsible flexible OFET array based on an organic semiconductor containing a photo-isomerizable azo group and its device performance. Before judging quality of this paper, the reviewer felt difficulty in understanding the experimental results exactly because a condition for each experiment is not clearly described: for example, the reviewer wonders whether the samples after UV-irradiation were thermally annealed, which is very important to know what the new findings are in this paper. Therefore, the reviewer requests exact description particularly on the experimental condition including the main text and figure captions before describing review report.

Response: Thank you very much for your comments, and we apologize for missing essential details in the manuscript. In this revised manuscript, the experimental details are added to the paper for checking. In particular, all tested samples are thermally annealed above the liquid crystal's transition temperature for *in-situ* crystallization before the light treatment and electric characterization. While during the follow-up light treatment and electrical characterization, the sample are kept at ambient temperature without experiencing further thermal treatment (the sample remains solid during all discussed processes). We have supplemented the experimental details in the article, including the main text and the figure captions.

Details are shown as follows:

1. Fig. 1 caption is revised with a sentence "All the experiments are conducted with film samples annealed at 80 °C unless otherwise stated."
2. "All the experiments are conducted with film samples annealed at 80 °C unless otherwise stated." is added on line 99 of the main text.
3. "The device array is prepared by continuous evaporation of AZO-BTBT-8, Au and HfO₂ insulating layers by a set of aligned shadow masks without annealing." is added in the caption of Fig. 5.
4. The post-processing condition is added in the device fabrication section in SI as "The device was fabricated by the spin-coating of a solution of AZO-BTBT-8 in CHCl₃ (annealed at 80 °C for 30 min) and thermal evaporation of Au through a designed mask."
5. "To ensure the reliability of the experiments, unless otherwise stated, all film samples are annealed at 80 °C before being characterized and tested." is added in the "General methods" section on Page 6 of SI.

Reviewer #2:

I have read the manuscript "Light-responsive Self-Strained Organic Semiconductor for Large Flexible OFET Sensing Array" by Mingliang Li et al. (MS # NCOMMS-22-04735) submitted for the publication in Nature Communications.

In their manuscript the authors reported the synthesis and characterization of AZO-BTBT-8 organic semiconductor (OSC), which was applied in the fabrication of a

flexible OFET device with reversible response upon UV-Vis irradiation. Reversibility is gained thanks to the trans-cis isomerization of azo-pendants, which induces a reversible stress-strain deformation with a consequent increase of charge mobility. The claimed possibility of developing light responsive OSC was confirmed by the fabrication of a prototype with different masks. However, it is opinion of the referee that the most delicate point of the research design is if the small changes, induced upon irradiation, in the molecular packing could be the real cause of the observed effect (self-strain). As the same authors wrote, the effect should be limited to top of the thin layer (line 61), and:

1. the change in the height is low (Figure 2.b),
2. the shift in XRD peaks is very small (Figure 2.h),
3. and the current data show a large dispersion (Figure 2.i).

It is well known that the conductivity can change upon photoisomerization of azocompounds present in polymeric materials and can be used for the preparation of light-responsive large area devices. Consequently, before the publication in NatCom the authors should investigate a blank device, in which the AZO-BTBT-8 is not aligned (in such a way there should not be any evident self-strain) and verify the presence/absence of any effect.

Response: Thanks for the comments. We agree that it is essential to identify the origin of the change in electrical properties is indeed from the photoinduced mechanical strain instead from other effects. It is well-known that the photoexcited carrier may be trapped inside the organic layers, leading to a long-lasting photodoping effect even after the light is switched off. We tried to design a series of experiments to identify the photo-induced self-strain effect. In Fig. R1a and b (Fig. 2a and b), the height change's absolute value is small (3.11 nm to 2.85 nm). However, this change accounts for over 8% vertical stretching, which is very significant in the crystal lattice. We remeasured the XRD peak shift shown in revised Fig. R1c (Fig. 2h); this time, we selected a lower angle diffraction peak for 21.05° shifting to higher fields with d -value increasing from 4.093 to 4.073 Å. Since the XRD peak shift shows an overall averaged lattice shift, the relative shift is smaller than the AFM measurement, only corresponding to the surface molecule rearrangement. This is also proved that the photo-switching reaction is mainly on the surface where the steric constraints are weaker, leading to the photo strain effect.

As shown in our experiment (Fig. R1e to h), the film sample shows a substantial enhancement of mobility from $0.015 \pm 0.007 \text{ cm}^2 \text{ V}^{-1} \text{ s}^{-1}$ (Fig. R1e and f without UV irradiation) to $0.141 \pm 0.009 \text{ cm}^2 \text{ V}^{-1} \text{ s}^{-1}$ (Fig. R1g and h with UV irradiation), where we propose as strain effect. As the referee mentioned, it is well known that the conductivity of an organic semiconductor can be shifted by photoisomerization of Azo-compounds, which can be attributed to the self-doping or photogating effect. However, carrier mobility is a more direct indicator of molecular packing. As shown in our result,

we did not see a noticeable doping effect as the onset potential is almost unchanged with the UV treatment ($\sim 2\text{V}$), suggesting that the doping introduced by trans-cis AZO compound transformation may not be the main cause for increased current.

On the other hand, we designed another experiment to rule out the photogating effect by showing that only the absorption of AZO, not BTBT, can induce such enhanced mobility. In this new control experiment, the film is treated with visible light ($\lambda \geq 420\text{ nm}$, which does not lead to molecular isomerization and self-strain). As shown in Fig. R1d, the film samples (thickness: 800 nm) strongly absorb to $\lambda \geq 420\text{ nm}$. The test results are shown in Fig. S18 in SI, and Fig. S 18 and Fig. 4g are listed below for your convenience. From the results in Fig. R1j (Fig. S18), a very small photocurrent on the order of $10^{-2}\ \mu\text{A}$ is observed, and is negligible compared to the large photoresponse under UV in Fig. R1i (Fig. 4g). Based on this observation, we conclude that the much-enhanced photocurrent under UV in Fig. R1i (Fig. 4g), is mainly a result of enhanced mobility due to the photoinduced self-strain effect.

Fig. R1 | **a** (Fig. 2a), The AFM image of AZO-BTBT-8 film. **b** (Fig. 2b), *In-situ* height profile curves indicated by the white two-way arrow in **a**. **c** (Fig. 2h), Enlarged XRD patterns with 2theta from 20.0° to 22.5° from Fig. S10. **d** (Fig. S8), UV-Vis curve of 800 nm film samples on quartz. **e**, **f** and **g**, **h** (Fig. 4a, b and c, d) are the transfer and output curves of AZO-BTBT-8 before and after UV irradiation on flexible PET, respectively. The average mobility increased from $0.015 \pm 0.007 \text{ cm}^2 \text{ V}^{-1} \text{ s}^{-1}$ (**e** and **f** without UV irradiation) to $0.141 \pm 0.009 \text{ cm}^2 \text{ V}^{-1} \text{ s}^{-1}$ (**g** and **h** with UV irradiation) on over 30 individual devices. **i** (Fig. 4g), Time trace of I_{DS} for the same device showing the reversible photoswitching events under alternating UV and Vis irradiation. **j** (Fig. S18), Time trace of I_{DS} for the device under Vis irradiation. $V_{DS} = -25 \text{ V}$, $V_G = -25 \text{ V}$. UV = ultraviolet ($\lambda = 365 \text{ nm}$), Vis = visible ($\lambda \geq 420 \text{ nm}$).

In addition, in the manuscript should be present an adequate relevance of the liquid crystallinity of AZO-BTBT-8 and a deeper investigation (why there are only few Maltese crosses? Pictures samples with a planar alignment at different angle between polarizers are required, X-ray data for the mesophase, ...).

Response: Thanks for the comments and suggestion. We have supplemented the liquid crystal related tests for a deeper investigation and added the results in the article.

To demonstrate the reproducibility, we conducted Maltese cross observation as shown in Fig. R2b. During the incubation, the Maltese cross appeared only at the molecular droplet. However, the sample droplets were discontinuous and scattered sparsely, so the number of crosses appears to be small. XRD data for the mesophase incubated at 80 °C is shown in Fig. R2c. Sample pictures with a planar alignment at different angle between polarizers are added as Fig. S4 and they are placed as follows for your convenience.

Fig. R2 | **a** (Fig. S2), The time-lapsed liquid crystal incubated at 80 °C. OM: optical microscopy; POM: polarized optical microscopy. **b** (Fig. S3a), The Maltese crosses incubated at 80 °C. **c** (Fig. S3b), XRD pattern for the mesophase incubated at 80 °C.

Fig. S4 | OM and POM images of the as-cast and annealed samples. The right column shows the POM images of samples rotated by 45 degrees.

In addition:

1. Scheme 1: The reported synthesis route of AZO-BTBT-8 is not accurate;

Response: Thanks for the careful checking. The route in Scheme S1 has been revised as follows:

Scheme S1. The synthesis route of AZO-BTBT-8

2. SI Lines 152-154: please, check calculations for the 1st kinetics;

Response: Thanks for the checking. The calculations have been checked and revised as follows:

$$dB/dt = k_{UV}(A_0 - B)$$

Integration of the rate equation yields:

$$\ln \frac{A_0}{A_0 - B} = k_{UV}t$$

$$\text{or } B = A_0 - A_0 \exp(-k_{UV}t)$$

3. SI Line 161: the reported Xe equation gives the complement of the conversion from trans to cis, and x 100 is missing;

Response: Thanks for the checking. The equation is revised as follows:

$$x_e = \frac{A}{A_0} = [\exp(k_{UV}t)]^{-1} \times 100\%$$

4. Line 101: from Figure 2.a the surfaces does not look like continuous and flat;

Response: We have rewritten this part of the text to make it clearer: "As shown in Fig. 2a, the film of AZO-BTBT-8 shows good continuity and flatness with molecular step edges, which is suitable for electronic device fabrication."

In Fig. 2a and b (Fig. R3a and b), a stepped edge and a plain structure with uniform color can be observed, indicating an ordered structure, which is beneficial to the fabrication of electronic devices. Colors in AFM represent relative heights, so areas of consistent color show good continuity and flatness. On the other hand, AFM only detects the surface morphology, the stepped edge structure also implies a continuous ordered structure embedded inside the film. The similar morphologies have been extensively studied in organic electronics. In Fig. R3c-d, the materials all exhibit a step structure and a regional flatness, and achieve good performance in the final electronic devices. (Fig. R3c: Jun-ichi Hanna et al., *Nat. Commun.*, **2015**, *6*, 6828; Fig. R3d: Marta Mas-Torrent et al., *J. Mater. Chem. C*, **2021**, *9*, 7186; Fig. R3e: Tatsuo Hasegawa et al., *Chem. Mater.*, **2018**, *30*, 5050)

[Redacted]

Fig. R3 | a, The AFM image of AZO-BTBT-8 film. **b**, *In-situ* height profile curves indicated by the white two-way arrow in **a**. **c**, **d** and **e** are AFM images and *In-situ* height profiles from literatures. (**c**: Jun-ichi Hanna et al., *Nat. Commun.*, **2015**, *6*, 6828; **d**: Marta Mas-Torrent et al., *J. Mater. Chem. C*, **2021**, *9*, 7186; **e**: Tatsuo Hasegawa et al., *Chem. Mater.*, **2018**, *30*, 5050)

5. Please, check values for mobility reported in the text and in the Figure captions. In Figure 4.f are the reported mobilitites before or after UV irradiation?

Response: Thanks for the suggestion and careful checking. Since this experiment is to test the stability of the device on flexible substrate, for simplicity, we included the mobilities before the UV radiation. With UV treatment, the mobility is consistently and reversibly increased by ~10 times. We have modified the caption and the main text to express more clearly.

The main text on Line195 is revised as “Under UV irradiation, the device's carrier mobility rises from $0.015 \pm 0.007 \text{ cm}^2 \text{ V}^{-1} \text{ s}^{-1}$ (Fig. 4a-b) to $0.141 \pm 0.009 \text{ cm}^2 \text{ V}^{-1} \text{ s}^{-1}$ (Fig. 4c-d) with a fixed on/off ratio of 10^5 , showing an increase of 9.4 times on average due to optimized molecular stacking as described previously.”

Fig. 4 is revised as follows:

Fig. 4 | **a, b** and **c, d** are the transfer and output curves of AZO-BTBT-8 before and after UV irradiation on flexible PET, respectively. The average mobility increased from $0.015 \pm 0.007 \text{ cm}^2 \text{ V}^{-1} \text{ s}^{-1}$ (**a** and **b** without UV irradiation) to $0.141 \pm 0.009 \text{ cm}^2 \text{ V}^{-1} \text{ s}^{-1}$ (**c** and **d** with UV irradiation), on over 30 individual devices. The average on/off ratio is approximately 10^5 . $L = 30 \text{ }\mu\text{m}$ and $W = 130 \text{ }\mu\text{m}$. **e**, I_{DS} of the OFET device under various mechanical distortions. The inset shows the transfer curves of the same device

under different bending conditions. **f**, Endurance cycles show the device's stability on flexible PET substrate without UV irradiation. **g**, Time trace of I_{DS} for the same device showing the reversible photoswitching events under alternating UV and Vis irradiation. **h**, Comparison of I_{DS} attenuation at different temperatures and under Vis irradiation. $V_{DS} = -25$ V, $V_G = -25$ V. UV = ultraviolet ($\lambda = 365$ nm), Vis = visible ($\lambda \geq 420$ nm).

Minor comments:

1. There some typos both in the manuscript and SI: thoroughly, 1 ft \times 1 ft (these are the dimensions of the original PET substrate), Gpa, thermodyanmically, scanning, tempertures,...;

Response: Thanks for the reminder. The manuscript has been fully revised carefully and the mistakes have been corrected.

2. SI line 124: data cannot agree with literature if AZO-BTBT-8 is a new compound;

Response: Thanks again for the careful checking. What you mentioned is very important. Small molecule organic semiconductors usually do not have good mechanical properties, so there are few works for us to refer to. In this work, we low our tone by “estimating Poisson's ratio and elastic modulus”, and only calculated the mechanical distribution to prove our points. We did not compare the calculated results with the PET data to avoid unsolid conclusions. However, we are aware of the problem, and strive to find experienced collaborators, hoping to solve the problems in coming projects.

3. Caption of Figure S2: it is not clear the means of “growing direction”. Are the two transition temperatures really referred to crystal \square nematic \square isotropic phase transitions? Could be present a smectic arrangement at lower temperatures?

Response: Thanks. By "growing direction", we originally intended to express the diffusion direction of the region in Fig. S2 that is discolored due to the growth of the liquid crystal. However, this expression is not clear, and has been deleted.

As shown in Fig. R4a, the smectic phase should have a larger-scale layer-ordered structure. But according to the data we obtained so far (incubated at 80 °C, Fig. R4c), no diffraction peaks are found in the low-angle range of the XRD pattern, indicating no layer-ordered structure (Jun-ichi Hanna et al., *Nat. Commun.*, **2015**, 6, 6828; Roland Resel et al., *Chem. Mater.*, **2021**, 33, 1455; Yves Henri Geerts et al., *Mater. Chem. Front.*, **2021**, 5, 249). So, we are inclined to regard this mesophase as "nematic", which already suffices to select the post-processing temperature.

Though 80 °C has been a low temperature in the liquid crystal temperature range 72.45-102.1°C (Fig. R3 b), we still do not rule out the appearance of smectic phase at lower temperature, which is also reflected in the work of other asymmetric BTBT derivatives.

(Jun-ichi Hanna et al., *Nat. Commun.*, **2015**, *6*, 6828; Roland Resel et al., *Chem. Mater.* **2021**, *33*, 1455; Tatsuo Hasegawa et al., *Chem. Mater.* **2015**, *27*, 3809)

Fig. R4 | a, Schematics for nematic and smectic phase. b (Fig. S1b), DSC plots for AZO-BTBT-8. c, (Fig. S3b), XRD pattern for the mesophase incubated at 80 °C.

4. Some acronyms are not defined;

Response: Thanks for the reminder. We have checked main text and the SI to add the missing definitions of abbreviations.

5. Caption of Figure S7: there are too many plots than absorption wavelengths;

Response: Figures and captions have been rearranged and revised as follows:

Fig. S9 | UV-Vis absorption spectroscopic studies. The gradual transitions of UV-Vis absorption spectra under UV (a, c, e, g, i and k) and visible light (b, d, f, h, j and l) irradiation. The left plots are the UV-Vis absorption spectra, and the right plots are the intensity extraction curves at the analysis wavelength from the left UV-Vis absorption spectra.

6. What about the differences in absorption between plots in Figures S6 and S7?

Response: In order to show the influence of thickness on the distribution and intensity of UV-Vis absorption curves, we extract the UV-Vis absorption plots of pristine samples with different thicknesses in Fig. S7 into Fig. S6.

Reviewer #3:

The authors showed a very nice example where light-induced isomerization of a photochromic azobenzene based organic semiconductor (OSC) led to exceptional device performance enhancement and reversible mobility switching in solid-state devices. A large-scale flexible OFET device was also fabricated on a flexible substrate using such azobenzene as photoisomerization OSCs, showing good stability and reproducibility. Various characterizations of such phenomena are presented to draw a conclusion that such light-responsive self-strained OSC provides a new strategy for designing and developing light-responsive OSCs and corresponding functionalized OFETs. The results are very impressive, especially when it was realized in a large scale flexible array that are reversible, programable, and scalable.

The results are original and should be of broad interest to readers in the field and related fields. I would recommend acceptance after minor revision.

Response: Thank the reviewer for his/her comments.

In the meantime, I suggest that the authors should also consider the changes in molecular packing caused by the light-induced isomerization as one of the reasons for the electronic performance enhancement. The height decrease indicates a closer packing, which usually leads to better mobility for OSCs. How does the intermolecular packing distance changes according to the calculations? The results on GIXD diffraction pattern indicate more crystalline packing. Whether that's a direct result of strain, or a strong intermolecular interaction, or the combination of both factors needs more discussion.

Response: Thanks a lot for the comments and suggestions. The DFT calculated distances of two-molecule systems are shown in Fig. S14 as follows. We found that the average distances between conformations is *cis-cis* (3.254 Å) > *trans-trans* (3.129 Å) > *trans-cis* (3.104 Å), which is consistent with the tendency of binding energy calculation results. Light-induced isomerization could synergistically optimize molecular interaction with dense stacking in an ideal condition and ultimately improves device performance. However, for a crystalline material, the well packed molecule lattice resulted significant steric constraint, which prevented the isomerization of molecules. It is well known that inside the crystal lattice, the AZO isomerization is strongly suppressed, and special molecular design is needed allow such solid-state

isomerization (Yasuo Norikane et al., *Nat. Commun.*, **2015**, *6*, 7310; Hong Meng et al., *ACS Appl. Mater. Interfaces*, **2017**, *9*, 7305; Christopher J. Barrett et al., *Adv. Mater.* **2013**, *25*, 1796–1800; Juyoung Yoon et al., *Adv. Mater.*, **2021**, *33*, 2007290; Hari Nalwa, *Handbook of Surfaces and Interfaces of Materials*, 1st Edition, **2001**, Academic Press). As such, although the improved molecular packing due to isomerization could be one of the reasons for improved device performance, we believe the strain effect should be the more direct cause. We have added related discussion in the main text as follows in Page 11:

“In addition, the average distance between the *trans-trans* conformations (3.129 Å) in the DFT calculation is larger than that between *trans-cis* conformations (3.104 Å), which indicates light-induced isomerization also synergistically optimizes molecular interaction with dense stacking and ultimately improves device performance.”

Fig. S14 | Distance measurements between conformations in DFT calculations. The average distances between conformations is *cis-cis* (3.254 Å) > *trans-trans* (3.129 Å) > *trans-cis* (3.104 Å)

Moreover, we experimentally ruled out two other possible reasons for the enhanced conductivity during isomerization: self-doping and photogating effect.

As shown in Fig. R5, we did not see obvious doping effect as the onset potential is almost unchanged with the UV treatment (~2V), which suggest the doping as introduced by *trans-cis* AZO compound transformation may not be the main cause for increased current.

On the other hand, we try to rule out the photogating effect by showing that the only the absorption of AZO not BTBT can induce such enhanced mobility. As shown in Fig. R6a, the 800 nm film samples have strong absorption to $\lambda \geq 420 \text{ nm}$. In this new control

experiment, the film is treated with visible light ($\lambda \geq 420$ nm, which do not lead to molecular isomerization and self-strain). A very small photocurrent on the order of 10^{-2} μA is observed in Fig. R6b, and it is negligible compared to the large current increase under UV in Fig. R6c. Based on this observation, photoinduced self-strain effect is left to be the main source to the enhanced mobility.

Fig. R5 | **a, b** and **c, d** (Fig. 4a, b and c, d) are the transfer and output curves of AZO-BTBT-8 before and after UV irradiation on flexible PET, respectively. The average mobility increased from 0.015 ± 0.007 $\text{cm}^2 \text{V}^{-1} \text{s}^{-1}$ (**a** and **b** without UV irradiation) to 0.141 ± 0.009 $\text{cm}^2 \text{V}^{-1} \text{s}^{-1}$ (**c** and **d** with UV irradiation), on over 30 individual devices. $V_{\text{DS}} = -25$ V, $V_{\text{G}} = -25$ V. UV = ultraviolet ($\lambda = 365$ nm).

Fig. R6 | **a** (Fig. S8), UV-Vis curve of 800 nm film samples on quartz. **b** (Fig. S18), Time trace of I_{DS} for the device under Vis irradiation. **c** (Fig. 4g), Time trace of I_{DS} for the same device showing the reversible photoswitching events under alternating UV and Vis irradiation. $V_{\text{DS}} = -25$ V, $V_{\text{G}} = -25$ V. UV = ultraviolet ($\lambda = 365$ nm), Vis = visible ($\lambda \geq 420$ nm).

In addition, the following minor issues should also be corrected.

1. Please specify the digital numbers in Fig. 1a correspond to which energy level (DFT calculation, CV and UV of solution or the solid-state films?) in the figure caption.

Response: Thanks. The figure caption has been revised as follows.

Fig. 1 | **a**, Energy level diagram of *trans* and *cis* conformations of AZO-BTBT-8. The orbital values are obtained from DFT calculations (black solid line), CV, and UV–Vis absorptions measurements (red dashed line for solution samples and purple solid line for solid films). UV, ultraviolet ($\lambda = 365$ nm); Vis, visible ($\lambda \geq 420$ nm). **b**, Schematic cross-section of an OFET with AZO-BTBT-8 as the semiconductor layer.

2. Scale bar in Fig. 2a didn't seem to be right. Please double check.

Response: Thanks. Scale bar in Fig. 2a has been revised. It should be 500 nm.

3. The right axis $t_{1/2}$ in Fig. 2f was not mentioned or discussed in the main text or the figure caption.

Response: Definition and discussion have been added in the manuscript. Thanks.

4. Keyword “Calculated” should be added to the figure caption of Fig. 3c. And description on calculation methods, the aspect ratio of the film section should also be included in figure caption.

Response: Thanks for his/her reminder. Fig. 3c has been revised. Related details are added.

Fig. 3 | Theoretical calculations of DFT and film mechanics. **a**, DFT energy schematic diagram under UV irradiation. **b**, DFT molecular binding energy diagram. The insets are the corresponding packing structures. **c**, The calculated distributions of the strain and stress in the long-axis direction (*x*-axis) in the organic film. The white arrow indicates the direction of stress decay. The aspect ratio of the film section is simplified to 8000:800. Stress, the internal forces that neighbour particles of a continuous material exert on each other. Strain, the measure of the material deformation. Both stress and strain are calculated in the *x* dimension.

REVIEWER COMMENTS

Reviewer #1 (Remarks to the Author):

This paper describes enhanced mobility after UV-illumination in organic FETs fabricated with an organic semiconductor material having a photo-isomerizable moiety of azo group and its demonstration of availability by fabricating a photo-responsible FET array.

I agree that the results described in this paper are new in terms of those that no one has reported ever.

However, trans-cis photo-isomerization in azo compounds is well-known, and it is also well known that the photo-isomerization causes not only a change in color but also changes in various properties.

Therefore, it is quite natural that one can expect light-induced mobility change in organic semiconductor material having azo moiety after light-illumination, even though no attention is paid to photo-induced strain effect on organic crystals. In this point of view, I am suspicious of what is new in this study.

The authors claim that the enhanced mobility in the azo-BTBT-8 FETs after illumination is caused by photo-induced mechanical stress in the illuminated Azo-BTBT-8 film. The experimental results including simulated strain and stress in the Azo-BTBT-8 film seem to support it, but they are not direct evidence of the effect. In fact, it is quite difficult to show the direct evidence for it with experiments.

As for the device performance in photo-switchable FET devices, the mobility of 0.15 cm²/Vs and mobility contrast of about 10 is something interesting, but exposure energy of UV illumination (20mW/cm²×2hr or 100 microW/cm² × 20min) is very high, which is far from the energy to call attention as a functionalized device. I am afraid that this energy is very difficult to reduce because it depends on the photo-isomerization efficiency, which is often a big problem of azo compounds in their practical applications.

In summary, I do not recommend publication of this paper in this journal on the basis of its standard.

There are some comments and questions for experiments in this study, which may help understanding the present results and whose answers should be described in the text, as follows:

1. The authors should pay attention to the fact that the strain effect on inorganic crystalline semiconductors, whose carrier transport is described by band transport, is quite different in organic crystalline semiconductors. In fact, strain effect collapses beyond a limited film thickness in inorganic crystalline semiconductors. It is different from the present results.
2. Why the mobility of FETs on PET is so different from those on Si substrate?
3. What is the annealing time at 80°C?
4. How does the mobility change before and after UV-illumination without thermal annealing?
5. How does the direction of UV-illumination affect the mobility change after UV-illumination and thermal annealing?
6. The film thickness of azo-BTBT-8 films in FETs is very thick, 800nm, which is 10 times thicker than that in typical OFETs. Is that for accumulation of the stress?
7. The same question as No.6; Why does the thicker film give higher mobility in Fig.S17, even though the conduction channel is far from the illuminated surface and access resistance is higher?
8. How does the crystal structure of the film without UV-illumination change before and after thermal annealing at 80°C?
9. The mobility change should be investigated as a function of UV exposure energy, which provides us with availability and limitation of this kind of devices.
10. The photo-induced isomerization of cis-trans in azo compounds are well established research subject. There are a lot of review articles appropriate for reference, rather than the one the authors cited.
11. As for the references, appropriate references should be cited to support the description of line 65 to 68 in page 5 related to authors view point in this study.

Reviewer #2 (Remarks to the Author):

I have read the revised version of the manuscript "Light-responsive Self-Strained Organic Semiconductor for Large Flexible OFET Sensing Array" by Mingliang Li et al. (MS # NCOMMS-22-04735A). In the revised paper the authors have answered in a satisfactory manner to all comments and, consequently, the manuscript can be published in Nature Communications as it is.

Reviewer #3 (Remarks to the Author):

The revised manuscript is in a much better shape now. One of the key concerns about the mechanism for the photo induced performance enhancement is addressed now with detailed and well-designed control experiments. I think this manuscript illustrated one important finding that are usually being neglected in the field of organic optoelectronic semiconductors. The results are original and should be of broad interest to readers in the field and related fields. I would recommend acceptance.

In the meantime, quite a few Supplementary Figures and tables were not mentioned in the main text. It would help the readers to understand the manuscript better if those data in the Supporting information were described in the main text accordingly.

Listed below are the details of our responses to the referees' comments.

Reviewer #1:

This paper describes enhanced mobility after UV-illumination in organic FETs fabricated with an organic semiconductor material having a photo-isomerizable moiety of azo group and its demonstration of availability by fabricating a photo-responsive FET array.

I agree that the results described in this paper are new in terms of those that no one has reported ever.

However, trans-cis photo-isomerization in azo compounds is well-known, and it is also well known that the photo-isomerization causes not only a change in color but also changes in various properties. Therefore, it is quite natural that one can expect light-induced mobility change in organic semiconductor material having azo moiety after light-illumination, even though no attention is paid to photoinduced strain effect on organic crystals. In this point of view, I am suspicious of what is new in this study.

The authors claim that the enhanced mobility in the azo-BTBT-8 FETs after illumination is caused by photoinduced mechanical stress in the illuminated Azo-BTBT-8 film. The experimental results including simulated strain and stress in the Azo-BTBT-8 film seem to support it, but they are not direct evidence of the effect. In fact, it is quite difficult to show the direct evidence for it with experiments.

Response1: Thank this reviewer for his/her professional and rigorous comments. We found that some details were missed in the previous version of the manuscript, and we have added or corrected them in the revised manuscript this time.

We also agree that the photoisomerization of azobenzene is a well-studied system, while most reported works describe the phenomenon and directly put it into use, leaving the working mechanism out. From the mechanistic point of view, the azobenzene isomerization will induce a significant structural change, which is quite different from other photoisomerization molecules like diarylenes and spiropyrans. Particularly, as a highly crystalline material (as our system, which is heated to liquid crystal phase, before solidified), one would rationalize that it will be difficult for azobenzene to isomerize due to the steric hindrance effect, which makes us suspect that the self-strain effect is likely to be an important reason for the improvement in device performance. Meanwhile, we also agree that it could be possible that this self-strain effect could also be at least partially responsible for the photo effect in some previous azobenzene-related work, while the mechanism has not been discussed in detail.

As the self-strain effect is difficult to demonstrate directly by experiments as the reviewer mentioned, we use UV, GIXD, and AFM in **Fig. 2** to support our claim by studying changes in molecular structure and packing arrangement, as well as DFT and stress/strain distribution simulation to help verify theoretical feasibility and round out the results, which is the main point of this paper.

Furthermore, we would like to express our great appreciation for the suggestion about the illumination direction effect. Following this suggestion, the angular dependence

photocurrent measurement is performed, which directly supports our conclusion that the observed device performance change is indeed due to the strain effect rather than directly resulting from the photoisomerization of the azo moiety. Please check the related explanation in Response 8.

As for the device performance in photoswitchable FET devices, the mobility of 0.15 cm²/Vs and mobility contrast of about 10 is something interesting, but exposure energy of UV illumination (20mW/cm²x2hr or 100 microW/cm² x 20min) is very high, which is far from the energy to call attention as a functionalized device. I am afraid that this energy is very difficult to reduce because it depends on the photo-isomerization efficiency, which is often a big problem of azo compounds in their practical applications. In summary, I do not recommend publication of this paper in this journal on the basis of its standard.

Response2: Thanks to this reviewer. We indeed missed to investigate the effect of UV intensity, and we have supplemented the relevant experiments in the text. According to the survey in **Table R1**, 100 μW cm⁻² is not significantly lower than previous azobenzene-related research (while still in the low end). In our experiment, to ensure complete molecular conversion and make the height change from photo-isomerization clearer, a higher power of 20 mW cm⁻² is adopted in the AFM tests in **Fig. 2**.

On the other hand, we do not want to claim that the high photosensitivity of our device is the major advancement in this paper. Instead, we expect that this newly proposed photoinduced self-strain effect may lead to a new method for better functional materials and devices of azobenzene compounds or provide some experiences for the research and application of other related materials.

Table R1. Survey of the UV intensities in azobenzene-related works

NO.	Literature	Citation	UV Intensity	Vis Intensity
1	Highly Sensitive Ultraviolet Light Sensor Based on Photoactive Organic Gate Dielectrics with an Azobenzene Derivative	J. Phys. Chem. C 2016 , 120 , 23172–23179	0.5–4 mW cm ⁻²	10 mW cm ⁻²
2	Structure Dependence of Photochromism and Thermo-chromism of Azobenzene-functionalized Polythiophenes	Express Polym. Lett. 2007 , 1 , 450–455	200–500 W	
3	A More Than Six Orders of Magnitude UV-Responsive Organic Field-Effect Transistor Utilizing a Benzothiophene Semiconductor and Disperse Red 1 for Enhanced Charge Separation	Adv. Mater. 2015 , 27 , 228–233	4.0 ± 0.5 mW cm ⁻²	
4	Thermal and Optical Modulation of the Carrier Mobility in OTFTs Based on an Azo-anthracene Liquid Crystal Organic Semiconductor	ACS Appl. Mater. Interfaces 2017 , 9 , 7305–7314	6 W	
5	Electric Bistability Induced by Incorporating Self-Assembled Monolayers/aggregated Clusters of Azobenzene Derivatives in Pentacene-Based Thin-Film Transistors	ACS Appl. Mater. Interfaces 2012 , 4 , 5483–5491	4 mW cm ⁻²	
6	Consideration of Azobenzene-Based Self-Assembled Monolayer Deposition Conditions for Maximizing Optoelectronic Switching Performances	Chem. Mater. 2021 , 33 , 5991–6002	50 mW	50 mW
7	Well-Defined Pillararene-Based Azobenzene Liquid Crystalline Photoresponsive Materials and Their Thin Films with Photomodulated Surfaces	Adv. Funct. Mater. 2015 , 25 , 3571–3580	500 W with filters (365 or 450 nm)	
8	Optically Tunable Field Effect Transistors with Conjugated Polymer Entailing Azobenzene Groups in the Side Chains	Adv. Funct. Mater. 2019 , 29 , 1807176	38 mW cm ⁻²	30 mW cm ⁻²
9	Reversible photoinduced bi-state polymer solar cells based on fullerene derivatives with azobenzene groups	Org. Electron. 2015 , 23 , 1–4	2.5 W cm ⁻²	100 mW cm ⁻²
10	Phototropic Discrimination of the Polymerization Behavior of Diacetylene Langmuir-Blodgett Films by an Azobenzene-Containing Monolayer	Adv. Mater. 1997 , 9 , 561–563	300 W	
11	Photoisomerizable Azobenzene Star-shaped Liquid Crystals: by Passing the Absence of Hydrogen Bonding	New J. Chem. 2022 , 46 , 7334–7345	5 mW	

There are some comments and questions for experiments in this study, which may help understanding the present results and whose answers should be described in the text, as follows:

Response3: Thanks for your comments. We have added the experiments and revised the manuscript accordingly. Please refer to the responses as following.

1. The authors should pay attention to the fact that the strain effect on inorganic crystalline semiconductors, whose carrier transport is described by band transport, is quite different in organic crystalline semiconductors. In fact, strain effect collapses beyond a limited film thickness in inorganic crystalline semiconductors. It is different from the present results.

Response4: Thank this reviewer for his/her reminder. It is a very good point to compare the strain effects between organic and inorganic semiconductors. As the reviewer mentioned, the material structure and the working principles of organic and inorganic materials are quite different. Strain effect collapses beyond a limited film thickness in inorganic crystalline semiconductors, such as doped Si, SiGe. However, we did not observe the similar result in AZO-BTBT-8 (as a typical organic material) for the following two reasons:

1. Structure determines properties.

Inorganic semiconductors (**Fig. R1a**) are usually composed of atoms bound together by covalent or ionic bonds and therefore have a high strength and modulus. The organic semiconductors have strong intramolecular covalent bonds and weak intermolecular van der Waals forces or π - π interaction. As illustrated in **Fig. R1b**, it is easy for them to slip and dislocate due to the weak intermolecular interactions (**Table R2**) and the large distance, resulting in flexibility and low modulus. So, inorganic materials will generate

more stress under the same strains, as a result of their high modulus (**Fig. R1c** and **d**).

[Redacted]

Fig. R1 | **a**, Structure of crystalline silicon (as a typical inorganic semiconductor); **b**, Common packing modes of small organic semiconductors. **c**, Schematic stress–strain curves for typical materials. **d**, Flexibility figure of merit for materials. (**b**: Wenping Hu et al., *Chem. Rev.* **2012**, *112*, 2208–2267; **c**: Xun shi et al., *InfoMat* **2021**, *3*, 22-35; **d**: G. Jeffrey Snyder et al., *Science* **2019**, *366*, 690-691)

Table R2. Energy table for different interactions*

Interaction	Energy range (kJ/mol)
Ionic bonds	700-4000
Covalent bonds	200-1000
π - π interactions	0-40
Hydrogen bonds	10-40
Van der Waals	0.4-4
Hydrophobic Interactions	<40

*Source: 1. <https://science.jrank.org/pages/984/Bond-Energy.html>; 2. [https://chem.libretexts.org/Bookshelves/Physical_and_Theoretical_Chemistry_Textbook_Maps/Supplemental_Modules_\(Physical_and_Theoretical_Chemistry\)/Physical_Properties_of_Matter/Atomic_and_Molecular_Properties/Intermolecular_Forces/Specific_Interactions/Van_Der_Waals_Interactions](https://chem.libretexts.org/Bookshelves/Physical_and_Theoretical_Chemistry_Textbook_Maps/Supplemental_Modules_(Physical_and_Theoretical_Chemistry)/Physical_Properties_of_Matter/Atomic_and_Molecular_Properties/Intermolecular_Forces/Specific_Interactions/Van_Der_Waals_Interactions)

2. Different fabrication methods.

In the epitaxial growth of inorganic semiconductors, mismatch results in strain and stress, and as the material expands into the third dimension, the stress grows (**Fig. R2a**, Paul K. C hub et al., *Mat. Sci. Semicon. Proc.* **2004**, *7*, 393-397; A. Borghesi et al., *European Materials Research Society Symposia Proceedings*, North Holland, 1996; Mehmet C. Ozturk et al., *Appl. Phys. Lett.* **2006**, *89*, 202118). However, due to the high strength of covalent bonds, the resulting stress cannot be released until materials' destruction. Compared with inorganics, organic semiconductors are usually processed by solution methods, such as spin coating (**Fig. R2b**). Although the stress is initially generated due to the disordered molecular arrangement after the rapid evaporation of the solvent, the postprocessing, like thermal annealing, will rearrange the molecules in an optimized stacking, releasing the stress. On the other hand, due to the relatively mild conditions of current and vacuum required for the sublimation of carbon materials in thermal evaporation (another organic-semiconductor-preparation method in this work), the organic molecules are deposited slowly (ca. 0.1 Å/s) on the surface of the substrate, and the ideal packing can be obtained directly without potential stress. (Aye M. Moh et al., *Phys. Status Solidi A* **2018**, *215*, 1700862; Gilles Horowitz et al., *Adv. Mater.* **1998**, *10*, 365-377)

In conclusion, compared with inorganic semiconductor materials, different structures and processing methods determine different stress results in organic semiconductor films.

We have added related statements on Line 48 of Page 4 in the introduction part as following: “It's worth noticing that, due to differences in material structure, the strain effect shows fascinating distinctions in inorganic and organic semiconductors, which deserves further investigation.”

[Redacted]

Fig. R2 | **a**, Illustration of the three growth modes found during the growth of compound semiconductors with various degrees of strain. Frank–van der Merwe (two-dimensional, FM), the Stranski–Krastanow (two-dimensional then developing into island, SK), and the Vollmer–Weber (island or three-dimensional, VW) growth modes. **b**, Molecular packing optimization of small organic semiconductor after thermal annealing; **c**, Illustration of the thermal evaporation (left) and the growth of pentacene film. (**a**: A. Borghesi et al., *European Materials Research Society Symposia Proceedings*, North Holland, **1996**; **b**: Daoben Zhu et al., *J. Am. Chem. Soc.* **2013**, *135*, 2338-2349; **c**: Sung-Hoon Ahn et al. *Int. J. of Precis. Eng. and Manuf.-Green Tech.* **2016**, *3*, 397–421.)

2. Why the mobility of FETs on PET is so different from those on Si substrate?

Response5: Thanks for the reviewer’s question. Although in principle, the device mobility from flexible substrate should be identical to the silicon substrate, we found that in the related works, the carrier mobility on flexible substrates is usually lower than that on silicon substrates. (Xiaohong Zhang et al., *Adv. Funct. Mater.* **2019**, *29*, 1902494; Xuefeng Guo et al., *ACS Nano* **2016**, *10*, 436–445; Tzung-Fang Guo et al., *Adv. Mater.* **2011**, *23*, 4077–4081) The possible reasons are listed as following:

1. Surface defects.

Charge carriers transport at the interface between the dielectric layer and the semiconductor layer, according to the working principle of OFET (**Fig. R3**), therefore the interface condition will determine carrier mobility. According to **Table S1**, the roughness of the PET substrate is substantially higher than that of the silicon wafer, especially after the deposition of alumina with lower processing temperature for PET. This suggests more defects and scattering sites for charge carriers on the PET substrate, thus lowering mobility.

[Redacted]

Fig. R3 | Working principle of p-type (left) and n-type (right) OFET.
(Source: <https://veniversum.me/ofet/>)

Table S1. Surface conditions for different substrates

Substrate	Substrate Roughness	Temperatures (°C) for ALD Al ₂ O ₃	Al ₂ O ₃ Roughness
Si	0.23 nm	200	0.35 nm
ITO-coated PET	24.38 nm	70	32.59 nm

2. Device quality.

Adjusting device fabrication parameters to retain the flexibility of PET will have an impact on device quality. On the other hand, PET has a natural curvature, and the thermal effect during the device fabrication will exacerbate its bending, speeding up the aging of organic materials, including PET and organic semiconductors, lowering device quality, and hence lowering the carrier mobility.

We have revised the texts on Line 205 on Page 13 as following: “In comparison to the device on silicon (Supplementary Fig. 21 and Supplementary Table 6), the flexible device maintains the expected response with moderate performance loss for the rough and bending substrate (Supplementary Table 1).”

3. What is the annealing time at 80 °C?

Response6: The annealing time is 30 min as we described in the “Device Fabrication” section on Page 5 of the supporting information. We have added this annealing time information on Line 115 on Page 8 in the main text as well.

4. How does the mobility change before and after UV-illumination without thermal annealing?

Response7: Thank this reviewer for the reminder. We have added the related tests in the supporting information and have them compared as follows.

Fig. S21 | OFET properties on silicon substrates. The transfer (left) and output curves (right) of AZO-BTBT-8 under different conditions. The average on/off ratio is approximately 10^5 . $L = 30 \mu\text{m}$ and $W = 130 \mu\text{m}$.

Table S5. Mobilities of the OFETs on the silicon substrate in **Fig. S21**

OFET devices	Mobility ($\text{cm}^2 \text{V}^{-1} \text{s}^{-1}$)	
	Before UV irradiation	After UV irradiation
Without thermal annealing	0.0071 ± 0.001	0.19 ± 0.07
With thermal annealing	0.23 ± 0.08	3.80 ± 0.09

We found that the mobility of as-cast film without thermal annealing is also improved after UV irradiation. The strain effect generated by photo-isomerization makes the

molecules pack more regularly, which works similarly to thermal annealing. Descriptions are added on Line 207 of Page 13 in the main text as “Interestingly, the mobility of as-cast film is also improved after UV irradiation, which indicates that the strain effect generated by photo-isomerization packs the molecules more regularly, similarly to thermal annealing.”

5. How does the direction of UV-illumination affect the mobility change after UV-illumination and thermal annealing?

Response8: We would like to express our appreciation for this great suggestion, which provides further evidence to support our proposed photo-induced strain effect. The relevant tests have added the results in SI. As shown in Fig. S10, the film absorption obeys Lambert-Beer law as expected, which means stronger light intensity at the surface facing the light source. We further investigate the relationship between the photoresponse and the incident angle (Fig. S23) and found:

a. Although the change in optical path caused by the incident angle influences the UV intensity on the interface close to the dielectrics (the surface UV light intensity remains unchanged at this time), the device response does not vary significantly.

b. Weak UV ($100 \mu\text{W cm}^{-2}$, not enough intensity to penetrate the device) irradiates from the backside of the organic film hardly show any photoresponse, whereas the strong UV (20 mW cm^{-2} , enough light intensity to penetrate the device) shows reduced photoresponse compared to the front side radiation.

It is well known that for OFET devices, the carrier is mainly conducted through a thin layer close to the semiconductor/dielectric interface. As shown in Fig. S23a, if the observed photoresponse is directly due to the photoisomerization of azo moiety, the front side illumination should result in a reduced photoeffect compared to backside illumination, where the isomerization is preferentially concentrated in conduction channel. Also, due to the different optical path length, the light intensity received at the semiconductor/dielectric interface also vary significantly with incident angle, which suggests a strong angular dependence. However, the experimental result shows otherwise: a. the angular dependence is not pronounced, b. the front side illumination is more important than the backside illumination. When the front side is not illuminated with enough UV (the weak UV from the back), almost no photoresponse is observed, while under strong UV from the back, which provides enough intensity after penetration to the front surface, the photoresponse is largely restored.

To explain this experimental observation, we conclude that the photoisomerization of the top surface is more important in the photoresponse of the OFET device, which can contribute to the device conductivity via strain effect.

Related descriptions are added on Line 212 of Page 13 in the main text as “The relationship between incident angle, and the light response has also been investigated. Due to the varied optical route, the device response is observed to be unaffected by the constant radiant intensity on the top surface and the various intensities on the bottom interface. On the other hand, front and back irradiation produce different responses to intense (20 mW cm^{-2} , enough light intensity to penetrate the device) and week ($100 \mu\text{W}$

cm⁻², not enough intensity to penetrate the device) UV light, respectively. Both results suggest that the photoisomerization of the top surface has a greater impact on the OFET device's photoresponse.”

Fig. S10 | Organic film UV absorption as a function of the UV intensity at 365 nm. Film absorption (A) = UV intensity absorbed by the film/ the incident UV intensity.

Fig. S23 | **a**, Schematic image for UV irradiation with different incident angles; **b**, The incident angle dependent response plot.

6. The film thickness of azo-BTBT-8 films in FETs is very thick, 800nm, which is 10 times thicker than that in typical OFETs. Is that for accumulation of the stress?

Response9: Thanks again for this comment. As we explained in Question 1, based on the loose structure of small crystalline organic molecules, the thermal-annealed spin-coated films and thermal-evaporated films hardly accumulate stress, which is independent of thickness. In our experiment, we utilized two methods for device fabrication: solution-based spin-coat, where we chose 800 nm thickness due to better film quality and reproducibility, and thermal evaporation, where 100 nm is applied for flexible array devices, where no significant performance change is observed. We investigated the thickness of organic semiconductors in related works in **Table R3**. Our organic semiconductor film is thick, but it is still within the reference range. On the

other hand, we want to clarify that the film thickness selection is mainly limited by our process rather than fundamental limitations.

Additional experimental details are added on page 5 of the SI regarding the device preparation steps as following: “200 nm Al₂O₃ thin film was deposited on PET by ALD at 70 °C for 2000 cycles. The device was fabricated by the spin-coating of a solution of AZO-BTBT-8 in 10 mg/mL CHCl₃ (annealed at 80 °C for 30 min) and thermal evaporation of Au through a designed mask. The device array on PET is prepared by continuous evaporation of AZO-BTBT-8, Au and HfO₂ insulating layers by a set of aligned shadow masks.”

Table R3. Film thickness survey in OFET devices.

NO.	Citation	Substrate	Organic Semiconductor	Thickness
1	Adv. Sci. 2017 , 4 , 1700007	Si	Pentacene	30 nm
2	ACS Nano 2016 , 10 , 436–445	PET	Pentacene	40 nm
3	Org. Electron. 2012 , 13 , 999–1003	Si	DE2	35–400 nm
4	J. Phys. Chem. Lett. 2020 , 11 , 1466–1472	Si	Pentacene	50 nm, 100 nm
5	Adv. Funct. Mater. 2004 , 14 , 811–815	Si	Pentacene	100 nm
6	Org. Electron. 2012 , 13 , 1614–1622	Si	TTF-TCNQ	150 nm
7	Appl. Phys. Lett. 2013 , 102 , 143301	Si	Pentacene	400 nm
8	Chem. Mater. 2007 , 19 , 4925–4932	Si	Thiophene-based star-shaped molecules	400 nm
9	J. Mater. Sci-mater. El. 2014 , 25 , 3727–3732	Si	Oligo(3-methylthiophenes)	700 nm
10	Adv. Mater. 2013 , 25 , 6219–6225	Si	C10-BTBT	800 nm

7. The same question as No.6; Why does the thicker film give higher mobility in Fig.S17, even though the conduction channel is far from the illuminated surface and access resistance is higher?

Response10:

[Redacted]

Fig. R5 | a, Carrier mobility as a function of the thickness of the pentacene film by thermal evaporation. **b (Fig. S19)**, Mobility curve of OFET device on silicon substrate with thickness dependence. (**a**: G. G. Malliaras et al., *Adv. Mater.* **2005**, *17*, 1795)

According to the OFET's working mechanism (**Fig. R3**), the charge carriers only transport in the semiconductor layer near the dielectric's surface. (G. G. Malliaras et al., *Adv. Mater.* **2005**, *17*, 1795; Ullrich Scherf et al., *Macromolecules* **2008**, *41*, 6800–6808) However, in practice, the thinner film is more difficult to be processed without defects and scattering sites. As a result, increasing film thickness within a certain thickness boosts mobility quickly, then slows down or stays constant after that. For our devices on the silicon wafers, the carrier mobility increases dramatically in the thickness range of 0–400 nm due to improved film quality.

8. How does the crystal structure of the film without UV-illumination change before

and after thermal annealing at 80 °C?

Response11: As demonstrated in **Fig. R2b** and **Fig. R6**, thermal annealing will help the amorphous film grow into well-packed crystalline films. It is also a common and well-studied post-processing method in organic electronics. (Concepcio Rovira et al., *Chem. Rev.* **2011**, *111*, 4833–4856)

We observe the crystal structure of the spin-coated films without UV-irradiation by polarized optical microscope, GIXD and XRD. As control experiments, the results are added to the SI. Under a polarized optical microscope, the crystal domains grew larger after thermal annealing, showing that regular molecular packing had formed inside the film. In GIXD tests (**Fig. R7**), the spinning-coated film went from having no signal to having diffraction spots and diffraction rings after annealing, suggesting the formation of large ordered structures inside the film. This is also supported by XRD studies (**Fig. R8**), which show that as-cast films exhibit only faint signals in the high field (30° to 50°), indicating that the films have a limited number of small-scale ordered structures. The enhanced high-field diffraction peaks indicate the increased number of the ordered structures after annealing, and the diffraction at 21.05° in the low-field, proves the larger-scale ordered structures inside the film.

[redacted]

Fig. R6 | **a**, AFM images showing crystal domain growing with annealing temperatures. **b**, XRD peaks move to lower field with annealing temperatures. **c**, Mobilities increase with annealing temperatures, especially for NDI3HU-DTYM2 (**a**: Hong Meng et al., *ACS Appl. Mater. Inter.* **2017**, *9*, 7305–7314; **b** and **c**: Daoben Zhu et al., *J. Am. Chem. Soc.* **2013**, *135*, 2338-2349)

Fig. S4 | OM and POM images of the as-cast and annealed film samples. The right column shows the POM images of film samples rotated by 45 degrees.

Fig. R7 | **a** (Fig. S11), GIXD diffraction pattern for the as-casted film samples; **b** (Fig. 2g), GIXD diffraction pattern sequentially for the pristine (thermal-annealed), UV-irradiated, and Vis-irradiated samples

Fig. R8 | **a** (Fig. S12) is XRD patterns of the film samples. **b** (Fig. 2h) is partially enlarged plots of **a**. The plots are in stack to show the peak shift.

9. The mobility change should be investigated as a function of UV exposure energy, which provides us with availability and limitation of this kind of devices.

Response12: The related tests have been done and added to the SI:

Fig. S22 | Carrier mobility on silicon substrate as a function of the UV intensity at 365 nm.

It can be seen from the experimental results that the carrier mobility has not reached the highest value with $100 \mu\text{W cm}^{-2}$ intensity, while the highest mobility is reached $\sim 5 \text{ mW cm}^{-2}$, and then the mobility decays slightly, which may be caused by the degradation of organic materials under intense UV irradiation. It can be seen that the photo-responsive devices based on azobenzene still face challenges such as slow response, poor stability, and low sensitivity. However, as the photoinduced strain effect decouples the conduction materials with photoresponse materials, better device performances can be expected by inducing new structures, such as bilayer organic layers and new materials, which should be investigated in the future.

Descriptions are added on Line 210 of Page 13 in the main text as “Furthermore, it is found that the device photoresponse increases with UV intensity at the low-intensity range (Supplementary Fig. 22). Intense irradiation, on the other hand, may result in molecular degradation against the device performance.”

10. The photoinduced isomerization of cis-trans in azo compounds are well established research subject. There are a lot of review articles appropriate for reference, rather than the one the authors cited.

Response13: Thanks for the reviewer’s advice. We have replaced the citation with some more literatures. Please refer to the following text:

Line 62 Page 4: “The BTBT backbone is alkylated on one end with octane to improve molecular flexibility and solubility,¹⁵ while the opposite end is covalently bound to azobenzene (AZO) to endow with reversible photoisomerization property.¹⁶”

Line 67 Page 5: “The AZO group in the as-synthesized AZO-BTBT-8 undergoes a switch from *trans* to *cis* conformation upon ultraviolet (UV) irradiation and back to the thermodynamically stable *trans* conformation with visible-light (Vis) excitation or high-temperature treatment.^{19, 20}”

References:

16. Kumar, G. S. & Neckers, D. C. Photochemistry of azobenzene-containing polymers. *Chemical Reviews*. **89**, 1915 (1989).
19. Wang, L. et al. High performance formaldehyde detection based on a novel copper (II) complex functionalized QCM gas sensor. *Sensors and Actuators B: Chemical* **248**, 820-828 (2017).
20. Volarić, J., Szymanski, W., Simeth, N.A. & Feringa, B.L. Molecular photoswitches in aqueous environments. *Chemical Society Reviews* **50**, 12377-12449 (2021).

11. As for the references, appropriate references should be cited to support the description of line 65 to 68 in page 5 related to authors view point in this study.

Response14: Literature references have been added, and please refer to the following text.

Line 67 Page 5: “Due to the steric hindrance^{16, 21-23}, most photo-inducing folding occurs on the top thin layer, in contrast to a fraction of the molecules inside the OSCs film. The vertical layer difference will induce uniform lattice strain to the bulk semiconductors, thus positively feedback to long-range ordered crystalline^{24, 25} and increasing the mobility.”

References:

16. Kumar, G. S. & Neckers, D. C. Photochemistry of azobenzene-containing polymers. *Chemical Reviews*. **89**, 1915 (1989).
21. Wang, Z. et al. Photoswitching in nanoporous, crystalline solids: an experimental and theoretical study for azobenzene linkers incorporated in MOFs. *Physical Chemistry Chemical Physics* **17**, 14582-14587 (2015).
22. Schweighauser, L., Strauss, M.A., Bellotto, S. & Wegner, H.A. Attraction or repulsion? London dispersion forces control azobenzene switches. *Angewandte Chemie International Edition* **54**, 13436-13439 (2015).
23. Valley, D.T., Onstott, M., Malyk, S. & Benderskii, A.V. Steric hindrance of photoswitching in self-assembled monolayers of azobenzene and alkane thiols. *Langmuir* **29**, 11623-11631 (2013).
24. Ichimura, K. Photoalignment of liquid-crystal systems. *Chemical Reviews* **100**, 1847-1874 (2000).
25. Bisoyi, H.K. & Li, Q. Light-driven liquid crystalline materials: From photoinduced phase transitions and property modulations to applications. *Chemical Reviews* **116**, 15089-15166 (2016).

Reviewer #2:

I have read the revised version of the manuscript “Light-responsive Self-Strained Organic Semiconductor for Large Flexible OFET Sensing Array” by Mingliang Li et al. (MS # NCOMMS-22-04735A). In the revised paper the authors have answered in a satisfactory manner to all comments and, consequently, the manuscript can be published in Nature Communications as it is.

Response: Thanks for the reviewer’s kind recommendation.

Reviewer #3:

The revised manuscript is in a much better shape now. One of the key concerns about the mechanism for the photo induced performance enhancement is addressed now with detailed and well-designed control experiments. I think this manuscript illustrated one important finding that are usually being neglected in the field of organic optoelectronic semiconductors. The results are original and should be of broad interest to readers in the field and related fields. I would recommend acceptance.

In the meantime, quite a few Supplementary Figures and tables were not mentioned in the main text. It would help the readers to understand the manuscript better if those data in the Supporting information were described in the main text accordingly.

Response: We have added the description of the unmentioned SI data in the main text. Thanks for the reviewer's kind recommendation.

Listed below are the major changes in the new version of the manuscript.

1. Explanation was added on line 48 of page 4 as “It’s worth noticing that, due to differences in material structure, the strain effect shows fascinating distinctions in inorganic and organic semiconductors, which deserves further investigation.”
2. Reference 16, 19, 21, 22, 23, 24 and 25 were added or modified. The other reference numbers were adjusted accordingly.
3. The missing figures and tables of SI were added in the main text.
4. Fig. 2h was modified.
5. Processing details were added on line 100, line 104, line 111, line 202 in the main text.
6. Explanation were added on line 205 as following: “In comparison to the device on silicon (Supplementary Fig. 21 and Supplementary Table 6), the flexible device maintains the expected response with moderate performance loss for the rough and bending substrate (Supplementary Table 1). Interestingly, the mobility of as-cast film is also improved after UV irradiation, which indicating that the strain effect generated by photo-isomerization packs the molecules more regularly, similarly to thermal annealing. Furthermore, it is found that the device photoresponse increases with UV intensity at the low intensity range (Supplementary Fig. 22). Intense irradiation, on the other hand, may result in molecular degradation, against the device performance. The relationship between incident angle and light response has also been investigated (Supplementary Fig. 23). Due to the varied optical route, the device response is observed to be unaffected by the constant radiant intensity on the upper surface and the various intensities on the lower surface. On the other hand, front and back irradiation produce different responses to intense (20 mW cm^{-2} , enough light intensity to penetrate the device) and week ($100 \mu\text{W cm}^{-2}$, not enough intensity to penetrate the device) UV light, respectively. Both of these results suggest that the photoisomerization of the top surface has a greater impact on the OFET device's photoresponse.”
7. Thickness of the AZO-BTBT-8 film by thermal evaporation was modified as 100 nm in the caption of Fig. 5 on line 235.
8. Processing details were added or modified in the sections of “Device fabrication” on page 5 and “General methods” on page 6 in SI.
9. Table S1, Fig. S10, Fig. S11, Fig. S22 and Fig. S23 were added.
10. The scale bar was added in Fig. S7.
11. Units in Fig. S9 were modified.
12. The plot for as-casted film was added in Fig. S12.
13. Transfer and output curves for unannealed films were added in Fig. S21. The results were summarized in Table S6.

REVIEWERS' COMMENTS

Reviewer #1 (Remarks to the Author):

The authors have answered my questions and comments for the previous manuscript seriously, providing additional experimental results. I appreciate their serious effort for them, and I am happy to know that my comments and questions helped to understand the mechanism in photo-induced mobility enhancement in UV-illuminated organic semiconductor thin films with azo-moiety. I think this revised manuscript is ready for publication as a paper with minor revision as indicated below, but I am still suspicious of the quality of the present paper in terms of originality worth publishing in this journal, as discussed below:

I looked at recent literatures in order to update my knowledge concerning photo-induced property change in azo compound, and found an interesting paper reported by Chinese research group in 2017, "Thermal and Optical Modulation of the Carrier Mobility in OFETs Based on an Azo-anthracene Liquid Crystal Organic Semiconductor, ACS Appl. Mater. Interfaces, 2017, 9, 7305-7314. This paper describes photo-induced mobility enhancement in an azoanthracene derivative, and almost same experimental results as the authors': anthracene is used for a semiconductive core moiety and decyloxyphenyl-azo group is attached to it to induce photo-isomerization in this paper, which is almost same strategy of molecular design as the Authors' basically, including a fact that the material exhibits liquid crystalline nature; in addition, experimental procedures including UV-illumination and post thermal annealing are the same as the authors'. Therefore, the results are basically the same as well, even though the present authors describe additional data for supporting their thought on the origin of mobility enhancement after UV-illumination and its device application.

The authors in the previous paper attributed the mobility enhancement after UV-illumination to "the synergistic effects of photo-isomerization and photo-induced molecular arrangement" on the basis of experimental results on X-ray diffraction and AFM studies.

I mentioned that "it is quite natural that one can expect light-induced mobility change in organic semiconductor material having azo moiety after light-illumination, even though no attention is paid to photo-induced strain effect on organic crystals" in the previous review report. In fact, the original idea of the photo-induced mobility change in Azo-containing organic semiconductor material has been realized in the previous paper published in 2017, as described above.

In conclusion, I hesitate to say that I do not agree to publish this paper in this journal on the basis of the criteria of this journal within my understanding for it.

There is a comment for further improvement of this manuscript.

Experimental conditions and explanation for the figures are not described clearly, for example,

1. Film absorption in Fig.S10 is difficult to understand what it means, even though Film absorption (A) is defined by UV intensity absorbed by the film/the incident UV intensity, because light absorption of a film does not depend on the incident light intensity, which is determined by absorption coefficient and film thickness.
2. There is no indication for I_{UV} in Fig.S23 (b): what does it mean?; is it I_{UV} after UV illumination and thermal annealing at 80°C for 30 min?
3. Does mobility in Fig.S22 mean the mobility in the FET after UV illumination and post thermal annealing at 80°C for 30min? What is illumination time, 30 min?
4. Please notice that the effect caused by light illumination usually depends not on the light intensity but exposure energy (light intensity x illumination time), as far as it is caused by the bulk properties of a material. The experimental results in Fig.S23 (b) and Fig. S22 do not follow the reciprocity law, as far as the exposure energy was kept constant in each experiment. I think that it may be a good sign that the present photo-induced change is not the case of the bulk effect in the materials, probably due to the interface effect as the authors expect, if so.

Listed below are the details of our responses to the referee's comments.

Reviewer #1:

The authors have answered my questions and comments for the previous manuscript seriously, providing additional experimental results. I appreciate their serious effort for them, and I am happy to know that my comments and questions helped to understand the mechanism in photo-induced mobility enhancement in UV-illuminated organic semiconductor thin films with azo-moiety.

I think this revised manuscript is ready for publication as a paper with minor revision as indicated below, but I am still suspicious of the quality of the present paper in terms of originality worth publishing in this journal, as discussed below:

I looked at recent literatures in order to update my knowledge concerning photo-induced property change in azo compound, and found an interesting paper reported by Chinese research group in 2017, "Thermal and Optical Modulation of the Carrier Mobility in OFETs Based on an Azo-anthracene Liquid Crystal Organic Semiconductor, ACS Appl. Mater. Interfaces, 2017, 9, 7305-7314. This paper describes photo-induced mobility enhancement in an azoanthracene derivative, and almost same experimental results as the authors': anthracene is used for a semiconductive core moiety and decyloxyphenyl-azo group is attached to it to induce photo-isomerization in this paper, which is almost same strategy of molecular design as the Authors' basically, including a fact that the material exhibits liquid crystalline nature; in addition, experimental procedures including UV-illumination and post thermal annealing are the same as the authors'. Therefore, the results are basically the same as well, even though the present authors describe additional data for supporting their thought on the origin of mobility enhancement after UV-illumination and its device application.

The authors in the previous paper attributed the mobility enhancement after UV-illumination to "the synergistic effects of photo-isomerization and photo-induced molecular arrangement" on the basis of experimental results on X-ray diffraction and AFM studies.

I mentioned that "it is quite natural that one can expect light-induced mobility change in organic semiconductor material having azo moiety after light-illumination, even though no attention is paid to photo-induced strain effect on organic crystals" in the previous review report. In fact, the original idea of the photo-induced mobility change in Azo-containing organic semiconductor material has been realized in the previous paper published in 2017, as described above.

In conclusion, I hesitate to say that I do not agree to publish this paper in this journal on the basis of the criteria of this journal within my understanding for it.

Response1:

Thanks to the reviewer for the recognition. We really admire the serious and rigorous attitude of the reviewers, which has also significantly improved our manuscripts in this review process.

We have carefully studied the articles mentioned by the reviewers. Although there are similarities in the molecular design, the materials we designed have shown more advantages, such as the use of BTBT with higher mobility. As mentioned by the

reviewer, it is logical that the addition of other moieties, such as azobenzene and alkyl chains, will lead to photoisomerization and liquid crystal properties. However, compared with the previous work mechanism, which was only mentioned in one sentence or usually ignored, we try to give a reasonable explanation to this well-known phenomenon in our research. As we argued in a previous rebuttal letter, we also believe the strain effect in photoswitchable organic semiconductors may be quite universal rather than unique. It is quite possible that the same effect had been at least partially observed previously (e.g., the paper found). On the other hand, our paper gave more direct evidence and discussed the mechanism in more detail, which is the novelty of this paper, not only limited to the molecular design and device performances. We have added in our citation as new Reference 17.

In addition, we also apply the designed materials to large-scale array sensing tests, hoping to provide some experience for the application of organic sensing materials.

There is a comment for further improvement of this manuscript.

Experimental conditions and explanation for the figures are not described clearly, for example,

1. Film absorption in Fig.S10 is difficult to understand what it means, even though Film absorption (A) is defined by UV intensity absorbed by the film/the incident UV intensity, because light absorption of a film does not depend on the incident light intensity, which is determined by absorption coefficient and film thickness.

Response2:

Thanks for the reviewer's rigorous reminder. In order to avoid ambiguity, we have removed the expressions related to absorption (A), and Fig. S10 is modified as follows:

Supplementary Figure 11 | $\Delta I/I_0$ as a function of the UV intensity at 365 nm. ΔI , UV intensity absorbed by the film; I_0 , the incident UV intensity.

2. There is no indication for I in Fig.S23 (b): what does it mean?; is it I after UV

illumination and thermal; annealing at 80C for 30 min?

Response3:

Thanks to the reviewer for the reminder. To avoid repetition, we have supplemented the experimental details in the method section as “To ensure the reliability of the experiments, unless otherwise stated, all film spinning-coated samples are annealed at 80°C for 30 min before being characterized and tested.”

Moreover, we added the definition of I in the figure caption as follows:

Supplementary Figure 24 | The angular-dependent photoresponse experiments. **a**, Schematic image for UV irradiation with different incident angles; **b**, The incident angle dependent response plot. I_0 and I_{UV} are current intensities before and after UV irradiation.

3. Does mobility in Fig.S22 mean the mobility in the FET after UV illumination and post thermal annealing at 80C for 30min? What is illumination time, 30 min?

Response4:

The mobility in the FET is after UV illumination and post thermal annealing at 80°C for 30min, as we mentioned in the method section: “To ensure the reliability of the experiments, unless otherwise stated, all film spinning-coated samples are annealed at 80°C for 30 min before being characterized and tested.”

The illumination time is 20 min, and this detail has been added to the figure caption as follows:

Supplementary Figure 23 | Carrier mobility on silicon substrate as a function of the UV intensity at 365 nm for 20 min.

4. Please notice that the effect caused by light illumination usually depends not on the light intensity but exposure energy (light intensity x illumination time), as far as it is caused by the bulk properties of a material. The experimental results in Fig.S23 (b) and Fig. S22 do not follow the reciprocity law, as far as the exposure energy was kept constant in each experiment. I think that it may be a good sign that the present photo-induced change is not the case of the bulk effect in the materials, probably due to the interface effect as the authors expect, if so.

Response4:

Thanks for the reviewer's detailed explanation. Indeed, the exposure dosage is a more direct cause for the photo effect, on the other hand, since our exposure time is long and the thermal relaxation of the photoswitched molecule is thus not negligible, which may also cause the deviation from reciprocal law. We believe the discussion in more detail will be too tedious and complicated, thus not included in this paper.

Listed below are the major changes in the new version of the manuscript.

1. Fig. S10 was revised as follows:

Fig. S10 | $\Delta I/I_0$ as a function of the UV intensity at 365 nm. ΔI , UV intensity absorbed by the film; I_0 , the incident UV intensity.

2. Annealing time was added on Page 6 of SI

To ensure the reliability of the experiments, unless otherwise stated, all film spinning-coated samples are annealed at 80°C for 30 min before being characterized and tested.

3. Fig. S23 was revised.

Fig. S23 | **a**, Schematic image for UV irradiation with different incident angles; **b**, The incident angle dependent response plot. I_0 and I_{UV} are current intensities before and after UV irradiation.

4. The illumination time is 20 min, and this detail has been added into the figure caption as following:

Fig. S22 | Carrier mobility on silicon substrate as a function of the UV intensity at 365 nm for 20 min.

5. Ref 17 was added.
6. Funding information were modified.
7. The affiliations were modified.
8. Fig. 2 was modified into 2 columns.
9. A short title for each part was added.
10. The method part was added.
11. The equations were numbered.
12. “Reasonable” was removed from Data availability.